# A Deconstruction of the Cross and the Crescent for Inclusive Religious Pluralism between Muslims and Christians in Nigeria

Ilesanmi G. Ajibola

Department of Christian Religious Studies, Federal College of Education, Zaria 810241, Kaduna State, Nigeria; gabajibola@hotmail.com

**Abstract:** The Crescent and the Cross as religious symbols are beyond the mere signification of religious affiliations. They are symbols on which over two hundred years of wars were sustained and are indicative of the religious dichotomy between modern Christianity and Islam across the globe. However, the tangential relationship between the usage of the symbols and the Jihad or the Crusade reeks of memories of fierce competition and unhealed historical memories. The collective memory of the wars fought under the symbols has remained a moniker for conquests and forceful submission. The exclusive propensities of the experiences are evident in the interreligious relation of the adherents of both religions in Nigeria. This article attempts to answer the question of how the exclusive religious disposition underlining most instances of religious crises in the country may be addressed. This article uses deconstructive analysis to strip the symbols and their exclusive religious dispositions for an inclusive religious pluralism model. It argues for the need for a critical rethinking of the exclusive interreligious model operative in the country to facilitate social development and the peaceful co-existence of the adherents of both religions.

**Keywords:** inclusive religious pluralism; Christianity; Islam; Nigeria

## 1. Introduction

Nigeria is the biggest economy in Africa and wields much economic and political influence in the West African sub-region. Nevertheless, weaponised religious interference in its internal social and political affairs has robbed the country of its developmental prospects. In 2023, Nigeria's general elections to the office of the President, governors, and the higher and lower houses of parliament take place. Electoral processes have commenced and are heated by the usual religious polarisation of political offices. Political permutations feature religious considerations along the line of Christian and Muslim differentiation. The ruling party's option of a Muslim—Muslim ticket (a Muslim and Muslim presidential and vice-presidential candidature) has witnessed resistance from Christians. The choice has been described as an insult which demonstrates a disregard for Christians. The scenario describes the often-polarised Nigeria polity along religious differences in political and social issues. The intrusion of religious bias in the social, economic, and political balance of the country's affairs has massively hampered its development. It has birthed antagonism, mutual suspicion, and fatal interreligious violence across the country. Against this background, there is no doubt that the evident religious bigotry needs to be addressed for any meaningful development to take place in the country. This article takes up the challenge and responds to the persistent antagonistic attitudes of Christians and Muslims towards each other. The article deploys the symbols of the Cross and the Crescent to depict the long history of religious conflict, particularly between Christians and Muslims in the country, as well as to represent the tension and violence between the two groups. The article suggests a deconstructive approach to overturn the hitherto exclusive interreligious

relation in the country with a continued interrogation of interreligious relation via inclusive religious pluralism.

## 2. Differentiated Religious Attitudes and Religious Exclusivism in Nigeria

Discourses on interreligious attitudes often feature three main strands: pluralism, exclusivism, and inclusivism. While religious pluralism holds that different religions can coexist in society and should be respected and accepted, religious exclusivism denies the privilege of the religious freedom to practice one's own religion without fear of discrimination or persecution. The third, religious inclusivism, recognises some truth in every religion and proposes that all religions lead to some form of a goal or ultimate reality. It must be noted, however, that inclusivism and pluralism do contain some similar trends, but they differ in content and quality. The former suggests that all religions have a common foundation and that the truth of one religion is not exclusive to that faith, but rather, such truth is present in other religions as well. The inclusivist's non-differential conclusion is not reached by religious pluralism, which argues for the freedom of religious independence, despite exclusive differences. The inclusivists propose an inclusive definition of their doctrines as the authentic version of similar truth in others' creedal expression. The controverted thoughts of Karl Rahner (1976) on *anonymous Christians* exemplify the theme of religious inclusivism.

Nigeria has witnessed several instances of religious crises arising from an attitude of religious exclusivism. Most of such instances are directly associated with using and applying religious symbols to situations that boomerang as religious protests. From the "Crisis over the Cross" conflict between Christians and Muslims in the south-west city of Ibadan, resulting in death and the destruction of properties at a very high magnitude (Hackett 1999, p. 555), to the exclusion of school pupils from admission into schools based on the acceptance or non-acceptance of symbolic religious wear (Lawal 2019; Nzwili 2021) exclusive religious discrimination against members of other religions has continued in the country.

The Federal Character Commission Act (No 34) of 1996, empowered by Sections 14 and 153(1c) of the Nigerian Constitution, emphasises the promoting, monitoring, and enforcing of compliance with the principles of the proportional sharing of all bureaucratic, economic, media, and political posts at all levels of government. Although this injunction might have been well-intended, it inadvertently promotes appointments based on ethnicity and religion in the country. Thus, there have been accusations of unofficial religious leaning by government officials at local, state, and federal levels in religious nepotism. For instance, it is common to read newspaper headlines such as, "Buhari's appointment has favored Hausas, Fulanis, Muslims more than Christians—CAN" (Yarima 2016) or as in the previous administration, "President Jonathan marginalized Muslims in ministerial and other appointments" (Hart 2015).

According to Adedeji (2015) the implementation of the Federal Character policy tends to consolidate the politicisation of ethnicity, with attendant religious polarisation. Hence, the Federal Character policy has argueably done great damage to Nigeria's secular status, which is not ideal for the stability of the polity and democratic sustenance. For instance, in the 2019 presidential campaign, some Christian candidates openly used the Cross as a symbol of their faith for campaigning among their likes, while some Muslim candidates used the Crescent as a symbol of their faith. In a context where religious symbols are used for campaigning, the idea is to give recognition to the dominant religion in the vicinity, at the detriment of the minority religious group. Of course, such campaigns are signals to the non-acceptance of the minority, and an indication that one religion was favoured over others at the location.

The exclusive religious tendencies among the religious adherents in Nigeria are often aggravated by political and religious leaders. The call for the implementation of Sharia law in the Muslim-majority northern states in 1999 was initiated by politicians and was seen by the Christians as an instrument through which Muslims intend to marginalise

and intimidate Christians in Muslim-dominated states (Uchendu 2020). There are also the words and actions of religious leaders who promote the idea that one religion is superior to others. For instance, Abubakar Gumi, an influential radical Islamic teacher in Nigeria, once argued, "by divorcing our government from God we are at once encouraging selfishness and unfounded ambitions" (Falola 1968, p. 128). Falola quoted Gumi to have emphasised:

> I have made several appeals before for a government founded on religion. Man is not a mindless animal whose only object in life is to eat, mate, sleep and die. Secularism, therefore, as the policy of operating government outside God's control, is alien to civilized human existence. We cannot expect to succeed in our affairs without abiding by the wishes of God, in spirit and in form.

It is important to note that the religion and theocentric governance to which Gumi refers is Islam. Gumi's position is not entirely different from what a Christian Clergy would say, given his strength and opportunity. The evident competition between Christians and Muslims in the country is not about to settle anytime soon, the reason being its inherent association with inherited colonial tendencies.

Although instances of oppositions to inclusive religious initiatives by the two religious' leaders are abound in Nigeria, elements of explorable initiatives by other leaders of both religions for alternatives to religious exclusivism are also evident. An exemplary instance is the non-violence and strategic engagement of two "once leaders of conflicting Christian and Muslim groups, (who) now work together to promote peace through the early detection and prevention of violence motivated by religious misunderstanding" (Interfaith Mediation Centre 2023). Both religious leaders had bitter experiences from the 1992 violent interreligious conflict where "the Imam and the Pastor were drawn into the fighting, and both paid a heavy price for their involvement—Imam Ashafa with the loss of two brothers and his teacher, Pastor James with the loss of his hand" (Center for Interreligious Understanding 2023). With over 10,000 members, the Centre is committed to a peaceful Nigeria where inclusive religious pluralism is plausible. Other examples are the Strength in Diversity Development Centre, Lagos, and the International Centre for Inter-Faith Peace and Harmony in Kaduna, Nigeria.

*An Inherited Imbalance of Religio-Political Dichotomy in Post-Colonial Nigeria*

The history of religious acrimony between Christians and Muslims in Nigeria is not new. For instance, (Lazarus and Button 2022) argued for an inherited colonial legacy as a lens to understand the continued tensed relationship between citizens in the country. According to Lazarus and Button, in the work mentioned above:

> since 1960, when Nigeria got her independence from the British government, northern politicians (e.g., Hausa and Fulani speakers) more often succeeded in ruling the nation than their southern (e.g., Yoruba and Igbo speakers) counterparts, partly due to the North's and South's differing experiences with imperial rule. Nonetheless, politicians of all stripes have been implicated in nepotism and corruption. Corruption and nepotism shape both licit and illicit distributions of the "national cake" (national wealth) by holders of political power and how elections are contested and won.

Tanko (1993) had proposed a similar argument as Lazarus and Button (2022) from a religious perspective. According to Tanko (1993, p. 120), shortly before and after Nigeria's independence from British colonial rule, the Muslim north was not comfortable with the growth of the Christian population in the north and took steps to curb the growth of such religious presence. The Christians in the north subsequently reacted. Presentto Aladeino in Tanko (1993, p. 120) noted that the formation of the National Christian Association (NCA) was a protest reaction against such steps, especially as typified in Sardauna's anti-Christian proliferation policies. To (Enwerem 1995), the rise of Islamic militancy in the north is not unconnected to this interest, where first, Muslims felt the need to liberate themselves from what they considered "the yoke of Euro-Christian values" imposed on the Muslim Ummah

(community) through the mediation of Western colonialism, and second, Muslims felt the need to recoup Nigeria's development through the centralisation of an Islamic worldview, such as that which existed during the time of Usman Dan Fodio. The claim of Enwerem on the need for Muslims to liberate themselves from the yoke of Euro-Christian values' has been further expressed in the weaponised resistance to Western education, as expressed in the founding philosophy of the Jama'atu Ahlus-Sunnah Lidda'Awati Wal Jihad group (Boko haram) (BBC 2016; Walker 2012).

The religious dichotomy is further compounded by the conviction in the superiority and distinctive claim to eternal truth by either religious body. While Muslims are deeply convinced of their religion and its precepts, they take their religion to be coherent and capable of aiding its adherents in navigating a world that is stained with "moral decadence, corrosion brought about by industrialization, and political errors" (Falola 1968, p. 31). Christians also see their religion and its precepts in the same light and are always on their guard to defend its fold from the marauding agenda of non-Christian evangelisers.

Furthermore, the religious constitution and demands on both Christianity and Islam to accord priority of place to mission ad extra (see Gospel of Mark chapter 16, verse 15, and the Holy Qur'an chapter 16 verse 125), make the sustenance of the Islamic religious conversion of others (*da'wah*) and Christian missionary activities imperative. Thus, holding the reins of power thus becomes a good opportunity to manipulate the state political mechanism for religious interest.

A relationship built on the exclusive claim of mutual religious rightness cannot but degenerate into mutual suspicion and a jealous guarding of religious creeds at the expense of other religions in the same vicinity. The spontaneous outburst of religious crises, as is often the case, betrays evidence of some form of bottled-up indignation. Few examples would suffice to bolster this claim; the October 1982 violent religious fracas in the northern city of Kano arose from an attempt by the Muslims to prevent the expansion of an existing decades-old Anglican House Church in the city. The reason was that expanding the church was considered a potential threat to the nearby, relatively new Mosque (Ibrahim 1989, pp. 65–66; Albert 2013). Another major incident which led to the loss of many students' lives and properties in the north-western city of Kafanchan in 1987, purportedly resulted from a Christian cleric quoting from the Holy Qur'an at a College of Education located in the town. Similarly, the "Crisis over the Cross" conflict between Christians and Muslims in the south-west city of Ibadan resulted from the request of the Muslims that a 42-year-old stone Cross, sited at the front of the Chapel of Resurrection at the University of Ibadan, be removed for the reason that the Muslims could "see the cross while worshipping" (Hackett 1999, p. 555).

Since mere triggers as minor as a personal altercation of a nonreligious nature in Secondary Schools (High Schools) could burst into huge clashes with religious dimensions, the reasons for such ensuing religious violence must have other motives. For example, in 1996, a violence with religious magnitude and the attendant destruction of properties began from a disciplinary incident between a Christian and a Muslim student at Government Science Secondary and Teachers College in Toro, Bauchi State; in 2005, there was a similar uprising between Christian and Muslim students in Kufena College, Wusasa, Zaria; and in 2006, another similar uprising occurred in Technical School, Malali Kaduna, and so on (Ajibola 2018a, p. 158).

The examples of exclusive religious dispositions identified above, and the struggle for the supremacy of religion in Nigeria, simultaneously develop along with a child's formative disposition in society. The correlation of these variables in a child's identity formation is upheld in Oppong's study on *Religion and Identity Development* (Oppong 2013). According to Oppong (2013), since religiosity is relevant in explaining "commitment and purposefulness in terms of identity formation, it is more likely that the strength of the relationship between religion and identity vary across different demographic groups as well as different epochs". Typically, the choice of a child's religious affiliation does not depend on the child's independent preference, but on an accident of birth and planned

parental directions. Thus, most Christians and Muslims became as such by virtue of being born into a Christian or Muslim family and confined to the dictates of the religions. Hence, traditioned from childhood, the strength of the linkage between religion and identity is well-internalised and informs attitude and action. The quality of the exclusive religious formation received from the family, as a microcosm of the larger society, gets back to society in a packaged interreligious dissonance. There must be an intentional effort at curbing the unproductive religious attitude that violates the freedom of the religious practice of the other.

So far, this article states that the experience of the tormenting interreligious relationships in the country results from saturated religious exclusivism. The same has been sustained by certain behavioural indices that Ajibola (2004) outlined to include the unreasoned assimilation of doctrines that could be inimical to peace. The indices must be deescalated through a disruptive deconstructionism to attain peace in the country. This article thus proposes a balance of pluralism and inclusivism, which is commonly referred to as inclusive pluralism. It is the belief that all religions are equally valid and should be respected and tolerated. In other words, the respectful recognition of the other's religion does not necessarily imply the same origin and goal. It is a religious disposition that is based on the idea that people should be able to practice their differentiated religions without fear of discrimination, despite exclusive beliefs. Imbibing such an attitude has the propensity to promote peaceful coexistence in society. It is often associated with the idea of religious diversity and the promotion of interfaith dialogue. Inclusive pluralism lays emphasis on the coexistence of different religious traditions and the recognition of the value and validity of diverse religious expressions through dialogue and mutual understanding.

## 3. Unhealed Violent Memories of Christianity and Islamic Religious Symbols

The Cross and the Crescent are religious symbols commonly associated with Christianity and Islam. The symbols became adopted as representing Christianity and Islam amidst unclear origins. Although neither of the symbols originated with the religions with which they are now identified, they have a huge secular and religious heritage with unsettling connotations. Ironically, both symbols evoke a sense of religious affiliation as well as collapsed trust between Christians and Muslims. Under the aegis of the symbols, inter-social bridges were broken through vicious religious propaganda and became the symbols of a carry-over of territorial-cum-political ambitions.

### 3.1. The Cross

The Cross is an important symbol in the Christian tradition. It is so important that the Roman Catholic Church, with a population of over 1.5 billion members worldwide, dedicated a feast for the Exaltation of the Holy Cross (September 14). The celebration commemorates the discovery of the True Cross and the dedication of the basilica and shrine built on the supposed site of the Crucified Christ.

While the use of the Cross for religious purposes predated Christianity, various forms of the pre-existing Cross have been adopted and used by Christians at various points in Christian history. For instance, the Coptic Orthodox Christians in Egypt adopted the ancient Egyptian *Tau* cross, also known as *crux ansata*, for Christian liturgical purposes. The *Tau* cross was originally an Egyptian hieroglyphic symbol of life, known as the *Ankh*. The various forms of pre-Christian Crosses, as used by different secular and religious groups, prefigure the adoption of the Cross as a symbol of the suffering and victory of Christ over death and evil forces. The association of the Cross with Christianity and Christians became immensely popular in the second half of the 4th century, with the conversion of Emperor Constantine. For the avoidance of doubt, the conventional commonplace Cross with a hanging Christ wearing a waist cloth and thorny wreath came a long way through non-religious history to religious adaptation.

Beyond the immense correlation of the Cross with Christianity from the 4th century and the various associated myths, the Cross, at a point in history, became a *raison d'état*

for military expeditions in the search for the true Cross on which Christ had died, and a platform to realise territorial ambitions. The latter reasons established a profound motif for mutual suspicion and distrust, especially in Christians' relationship with Muslims.

*3.2. The Crescent*

Certain Islamic religious holidays and festivals have been regulated on the phases of the moon cycles. For example, the beginning and end of Ramadan fasting are determined by the new moon's Crescent. Similarly, many Muslim countries have adopted the Crescent moon and star as their national symbol, both of which are prominently displayed on their country's flags. Iconic depictions of Islam with the Crescent and the star in popular Islamic arts and culture also abound. Yet, the Crescent and star as an Islamic religious symbol is neither official nor of Islamic origin.

The association of the Crescent and the star with Islam is often traced to the Ottoman Empire. According to Andrew Traver, the symbol was developed in the Greek colony of Byzantium ca. 300 BC (Traver 2002). However, the origin of the Crescent moon in religious ambience is shrouded in oblivion. What is clear, however, is that the symbol predates Islam. Traces of the practical use of the symbol with religious connotations existed in the Ancient Near East. Similarly, many ancient Greek (classical and Hellenistic) as well as Roman religious emblems with the stars and Crescent insignia have also been found (Faraone 2018). While there is no direct or concrete connection between the concept of Islam and the starred Crescent symbol, the association of its presence on the Ottoman flag, as circulated and adopted in the wide Ottoman Empire and caliphate, depicts it as a symbol of Islam by association. Of course, there are Muslims who, for the reason of no direct link with Islam, have rejected the symbol as Islamic.

However, there are two common denominators to the Cross and the Crescent, as the symbols represented the religions in their various capacities, and in association with bitter religious wars. In reference to the latter sense of usage, while the Christian fighters adorned their war apparel with the Cross and tagged the wearers as Crusaders, the Muslim fighters had the Crescent moon insignia.

## 4. Depleted Grandiose Symbols of the Cross and the Crescent

The transition of symbols from the original and intended meaning into besmirched versions of the same symbols occurs gradually. It is a movement that takes some time; so was the case with the Cross and the Crescent. Christians and Muslims obfuscated the use of the Cross and the Crescent in their historic struggle for land and religious conversion. The Cross became associated with "crusade" and the Crescent also became synonymous with Jihad. The symbols subsequently acquired religious profiling associated with conquests and forceful submission. In Nigeria, the impact of the besmirched versions of the concepts has lingered, and now represents indicators for disaffection and disunity among the adherents of the two prominent religions. The obfuscation of such images has remained an albatross to peace and development in the country.

According to Gopin (2000), religion and their narratives have the potential for double roles in violence and peacemaking. Gopin (2002), further connects the roles and power of myths, rituals, and dialogue as key indicators in violence and peacebuilding. The directly linked indicators of these variables in religious violence cannot be divorced from the individual identity formation index. In the famous *Identity, youth, and crisis* book by Erikson (1968) the connection between childhood identifications and identity formation in adolescence was established. Erikson's work and other similar research correlate identity formation and the growth impact index to include a variety of socio-cultural factors. According to a Y Studio (2020), some of the social factors that could impact identity formation include a "variety of internal and external factors like society, family, loved ones, ethnicity, race, culture, location, opportunities, media, interests, appearance, self-expression and life experiences". Therefore, an individual may rightly be described as a product of their formative society and the attendant variables. Notably, what makes

an individual is not expressed in isolation. The decision and expression of oneself affect even generations beyond the present. Hence, the Y Studio (2020) noted that personal and collective identities "evolve and influence the identities of future generations". It is deducible from Gopin and Erikson's works that our current collective religious experience is a function of societal formation. Stated bluntly, the Nigerian society did not arrive at the current tensed interreligious experiences by chance. Our collective experience of religious violence has been associated with an ubiquitous presence of deteriorating moral and religious values (Adedayo 2020; Akpan and Uyanga 2022), as well as prevalent exclusive religious dispositions (Ajibola 2018a). The conclusion of Ajibola is informed by his content analysis of the subsisting Christian Religious Education curriculum operative in Nigerian Colleges of Education, where Basic and Secondary School teachers are trained (Ajibola 2018a). In the same work, Ajibola finds out that the curriculum is deliberately made to be confessional. This claim is clearly made in the *Nigeria Certificate in Education Minimum Standards for Arts and Social Sciences Education* (Federal Republic of Nigeria 2012, pp. 10, 21). The document stated that the objectives of the Christian Religious Studies include expressing "accurate knowledge of God the Father, Son and Holy Spirit needed to live as a Christian in the community", and "radiate attitudes and values which are typical of a mature and responsible member of the Christian community" (Federal Republic of Nigeria 2012, p. 10). The same goes for Islamic Studies, which are meant to "understand Islam as a culture and civilization" and to "instill in the students the spirit of God consciousness, to lead them to appreciate and uphold the values and teachings of Islam, and to live by it" (Federal Republic of Nigeria 2012, p. 21).

Although Mosques and Churches adorn every corner of most Nigerian streets, thereby creating a notion that Nigerians are extremely religious, the confessional nature of the government-approved religious education curriculum has continued to dispose school pupils to religious exclusivism from a very young age. Consequently, exclusive religious disposition becomes so deep-rooted in the psyche of most citizens to the extent that interpersonal relation is regulated by an exclusive confessional daily order. In a religiously diverse society such as Nigeria, a curriculum that is rich in common themes in both religions should be prominent. Themes including the common religious ancestry, call to service, and emphasis on shared religious values, with examples from both religious scriptures, would go a long way in promoting inclusive religious pluralism.

Nigeria's religious demography loosely identifies particular geographical locations with certain religions. Coincidentally, the natural landscape demarcation of the north from the east and the west, now associated with religious identity, has continued to be a promotional ticket for religious exclusivism in the country. A report from Marshall (2020) for the Berkley Center for Religion, Peace & World Affairs observed a scenario where the violent attacks against Muslims in the south present a reason for the violent retaliation against Christians in the north. Deliberate and implied efforts, as reported by the Report from the Berkley Centre for Religion (Marshall 2020), kept sustaining the religious outlook of the exclusive religious demography of the country, which has continued. For instance, the religious education curriculum in Nigeria is deliberately tailored for evangelism, to promote the specific interest of the various religions. One of the panelists who drafted the current curriculum had argued that "the way of teaching Islam and Christianity in Nigeria is expected to be confessional, that is, students are taught how to practice their religion as well as being taught about their religion" (Lemu 2007, p. 222). Thus, the religions are systemically imbued in schools' religious curricula, and unfortunately made to be confessional. A confessional curriculum promotes the religious contents for which it is designed, at the expense of other religions. Products of such religious design may only be adept at preaching to convert others and promote an individual's religion or beliefs despite all odds. This is reminiscent of the medieval conquest mentality. Beyond children's immediate family exposure to their parents' religions, children are compelled to attend Islamiyah and catechism classes, respectively. The confessional nature of religious studies in Nigerian schools, and the race between Christians and Muslims to outsmart one another

in the social and political equation of the country, has led to mutual suspicion, distrust, and inhibited discontentment between the adherents of Islam and Christianity in the country (Ushe 2012; Ajibola 2018a).

## 5. A Deconstruction of the Cross and the Crescent for Religious Peace and Social Development in Nigeria

It must be stated from the outset that deconstructionism is a method of critical analysis that does not entail a rejection or devaluation of any symbol, but is an attempt to understand the underlying assumptions, biases, and power relations that are embedded in them. The Cross and the Crescent symbols are simultaneously reminiscent of unity and conflict. While the Christian Cross represents both Christ himself and Christianity, the stark memorial of its connection with the violent confrontation of Christian Europe on Muslim monarchs steers everyone frontally. Similarly, while the Crescent is not directly linked with the faith of Islam, its link with a series of religious wars with Christian Europe in the Ottoman Empire who ruled over the Muslim world at the time invokes a mixed memory of an ambitious conquest. Both symbols, therefore, are reminiscences of territorial invasion, conquest, and occupation based on the religious differences between Christians and Muslims. Nevertheless, the associated meaning of violence ascribed to these symbols is neither fixed nor static. Still, as evolving symbols adapted to the needs of both religions, they must be seen as constantly evolving.

The fluid origin and contextual usage of the Cross and the Crescent points to the non-static correlation of the symbols with neither Christianity nor Islam. Hence, it is possible to go beyond the residues of the unfortunate religious wars that were fought under the symbols' insignia to discover a reconciliatory meaning that would be contemporary with Nigeria's search for peaceful coexistence. In other words, there is no natural correlation between the Cross and the Crescent as they are now used in Christianity and Islam to describe incidents of violent invasion, conquest, and occupation, and the assumed desire of religious adherents to perpetrate such memories. Rather, a constant interrogation and negotiation with the historical signification of the symbols and the religions they signify would facilitate a healing function of the symbols.

To achieve the overturn of the carry-over of the associated meaning of the Cross and the Crescent as a symbolic representation of territorial ambitions, the deconstructive tradition of Jacque Derrida shall be considered. The choice of Derrida's deconstructionism for a study in overturning the burden of violence associated with the use of the Cross and the Crescent is from its potential as a framework for the critical examination and challenge of traditional interpretations. Derrida's deconstructionism challenges traditional interpretations of language, meaning, and symbols. By deconstructing the symbols, the inherent complexities of the symbols are revealed, and fixed meanings are questioned. Thus, such a framework could open possibilities for the reevaluation and recontextualisation of the Cross and the Crescent.

It is important to note that the so-called stages in Derrida's deconstructionism may not be found discussed in a manner that explicitly outlines distinct stages. Conceived as the first stage, and in relation to the theme of this article, is Derrida's focus on the internal contradictions and inconsistencies of signs, text, and concepts. In his famous book *Of Grammatology*, Derrida (1967, pp. 14–15) hints at the "hidden sediments" in "metaphysico-theological roots" being conceivable as religious symbols, and argues that such symbols are always open to multiple interpretations, and that no single interpretation can ever be definitive, but is inherently ambiguous and unstable. In the second stage, implicit in its hypothetical response to Heidegger's spirituality, a relationship between religious symbols and the social and political context in which they are used correlates religious symbols to the construction and maintenance of power relations (Derrida 1976, pp. 9–12). From such a stance, religious symbols are susceptible to oppressive and marginalising manipulations. Hence, through deconstruction, one can reveal the historical, cultural, and

linguistic contingencies that shape the understanding of symbols, thereby undermining their presumed stability and fixed meanings.

Summarily, associated meanings to concepts are defined equally by what an institution considers as its functioning meaning and what it does not (Derrida 1982b). The projection of meaning depends on the inscribed *differences* in the structure of meaning. Hence, the variation of the intended meaning by the originator of the symbols and the constrained meaning that emerged via interpretation oscillates according to the context of usage. At each point of the operation of the symbols, a particular usage outdoes the other and becomes dominant. That usage, even though it becomes dominant at that moment, does not reduce the totality of the symbol to that contextual usage. In fact, "while the idea of exclusion suggest(s) the absence of any presence of that which is excluded, . . . that which is instituted depends for its existence on what has been excluded" (Turner 2016). The carry-over of suspicion and the exclusive religious interaction between Muslims and Christians in Nigeria is not *prima facie,* but an accident in interreligious relations, and as such, it can be disrupted through a well-structured interreligious model.

It has been demonstrated in this article that the Cross and the Crescent have no fixed meaning. Thus, the unintended violent interpretation of their influences between Christians and Muslims in the 22nd century Nigeria needs to be overturned. The Crusades' Cross and the Jihad's Crescent belie the Christians' ideal of love, as preached by Jesus and the Muslims' Islam (as a religion of Peace). Christians and Muslims in Nigeria must move from a mere inversion of the symbols' outdated meanings to embrace the second stage of Derrida's deconstruction efforts, which attempt to strip undermined religious symbols of presumed stable and fixed meanings that may promote unhealthy profiling, marginalisation, and even oppression. In other words, Nigerians must continue to engage the symbols in the positive distilling and interrogation of the structure of their meaning for the mutual peaceful coexistence of citizens. This nuanced understanding of the role of religious symbols in society as both positive and negative is also in line with Derrida's deconstructionist view (Derrida 1982a, pp. 1–28). The argument portents that religious symbols can be used to promote violence, oppression, and exclusion, just as they could also be used to promote peace, justice, and inclusion.

It is thus proposed that in place of the exclusive religious disposition arising from the inhibited consequences of religious symbols of war and violence adorning our Churches and Mosques, an inclusive religious pluralism should be our interactive religious model. It is a model that is open to the religious other while recognising the validity of their religious rights and beliefs (Ajibola 2018a; Dupuis 1997, 1999; Iwuchukwu 2009; Phan 2003). In other words, 'I recognise and accept the sanctity of your religion and your right to your belief and practices, while holding unto mine'. To the Christians and the Muslims, such a disposition must be held in tandem with their religious obligations which encourage proselytism for conversion. The catch is that such an obligation can be duly conducted without necessarily engaging polemical media, but by simply living righteously and exemplarily. Such a model of interreligious dialogue is achievable in Nigeria, where according to Ajibola (2018a), "despite the difference in doctrinal and some emphasis on aspects of their varied religious traditions, Muslims and Christians' bodies have continued to work together on areas of common interests". These are explorable building blocks for peace and viable bases for inclusive religious pluralism.

## 6. Practical Steps in Deconstructing the Saturated Cross and Crescent's Effects

The many violent conflicts in Nigeria's religious space have bred unhealthy competition between Christians and Muslims. Such "uncompromising competition" as Iwuchukwu (2013) calls it, was witnessed in the ancient struggle for land between the Cross and the Crescent, but has led to mutual suspicion, increased excesses of religious extremists, and political blackmail of religion in the 22nd century Nigeria. A deliberate and well-calculated effort at a solution to the saturated tensed religious relationship must ensue from diffusing the symbols of associated obfuscation, and this is where deconstructionism comes in.

Simplistically, deconstructionism entails the fluid consideration of concepts rather than the ideally static and fixed understanding of such concepts and the medium of communication. For this study, it would mean an overthrow of fixed exclusive religious expressions by Christians and Muslims. The meaning of the Cross and the Crescent as love and peace must be placed in tandem with the opposing violent hierarchy of hate and war. To exclusively consider the Cross as a symbol of Christ's love or the Crescent as a symbol of the religion of peace is to unresolvedly foreclose the attendant meaning on which they were signs of contradictions. Similarly, to remain rigid in associating the meaning of the symbols to the experiences of the 11th to the 13th century is to be unprogressively fixated. Thus, the father of deconstructionism, Jacques Derrida, warned that a concept must be understood in the context of its opposite (Derrida 1982b). The religious exclusivism must be held in the light of the opposing pluralism, and pave way for negotiating an inclusive religious disposition.

The mid-way between outright religious pluralism and rigid exclusivism must be engaged in the continued negotiation of the religious, social, and political contract of Nigerians. Although Derrida's deconstructionism does not aim at a further reconstruction of inverted concepts, his position does not foreclose a progressive understanding of concepts as may strengthen the ideals of an institution. The proposed deconstruction of the Cross and the Crescent is not about Christianity or Islam, but the rigid structures within these institutions that are, or could constitute, obstacles to meaningful and impactful social development in the country. In Nigeria, the guarding of religious and associated beliefs is given assurance by the confessional-centric religious curriculum in use in the country's educational institutions. In other words, the religious education curricula in Nigerian schools are deliberately made to safeguard and promote the exclusive identity and dogma of Islam and Christianity (Lemu 2005; Ajibola 2018b). Such an arrangement cannot but breed mutual suspicion, unhealthy competition, and eventual degeneration into violent conflicts.

Given the popular confinement of the freedom of religious expression which is against the religious rights of citizens enshrined in the Nigeria Constitution (Federal Republic of Nigeria 1999, sec. 38), and the propagation and perpetuation of religious exclusivism via radicalising media such as radio, television, and worst of all, the religious studies curriculum from primary schools to tertiary institutions, the country may not have seen the end to interreligious conflict. Thus, there is an urgent need to undo the religious narratives encapsulating the negative ingress of the Cross and the Crescent. Nevertheless, the effort must not be seen as a one-time solution to the problem of religious crises emanating from an ambition for religious conversion or an inordinate desire for religious supremacy.

Since the past cannot be changed, but there is the possibility of engaging the present for a better future, the proposed model of inclusive religious pluralism must be fairly considered. In this direction, formal religious education could play a major role. As noted earlier, the present religious studies curriculum in Nigeria is confessional-centred. The curriculum needs be altered to accommodate a change in the religious attitude of the younger generation—the hope of the country's future. Similarly, since storytelling is a key Nigerian value for *traditioning*, the non-formal religious instructions in catechism and at other non-formal spaces must be intentionally set up to effect analytic thought patterns that accommodate religious openness, the readiness to learn from each other, and an acceptance beyond the mere tolerance of the religious other.

Furthermore, the role of religious leaders in effecting a positive change for inclusive religious pluralism cannot be overemphasised. The work of Dowd on *Christianity, Islam, and Liberal Democracy: Lessons from Sub-Saharan Africa* (Dowd 2015) has greater lessons than suggested by the title. Details of the research indicate that in a religiously diverse and integrated environment, religious leaders tend to be more encouraging of civic engagement, democracy, and religious liberty. Such evidence suggests a capacity to prompt policymakers to inspire support for the promotion of interreligious peacebuilding. As a country with a large Muslim and Christian population, religious tolerance and cooperation is imperative.

Hence, the presence of many Muslim-Christian schools and hospitals, and many interfaith organisations working to promote peace and understanding, could encourage religious leaders to work together to promote peace and understanding, speak out against violence and intolerance, support initiatives that bring people of different faiths together, such as community events, sports leagues, and educational programs, and make it easier for people of different faiths to interact with each other, such as by providing translation services and making government buildings accessible to people of all faiths. Furthermore, this article argued that identity formation plays an immense role in the understanding and disposition of a religious adherent towards a religious other. The proposed inclusive religious pluralistic formal and informal curricula must be infused with appropriate behavioural objectives that would impact desired openness to other religious adherents. Of course, such a curriculum must be implemented by trained teachers of interreligious studies and managed by sound interlocutors in an atmosphere of openness and sincerity of purpose. Similarly, a time frame for the assessment and reassessment of the gains of the curriculum outcomes needs to be stimulated and engaged, especially with the formal curriculum.

## 7. Conclusions

Religious symbols are strong kinetic representations of religious ideas and events, such as the Cross and the Crescent in the psychology of Christians and Muslims in Nigeria. The symbols simultaneously evoke opposing meanings and sentiments depending on which side of the fence one stays. Nevertheless, the divisionary signification of the symbols is perpetuated by the deliberate guarding of the exclusive religious education pattern in the formal government-regulated education curriculum in schools across the country. The confessional-based curriculum structure, and the one-sided apologetic religious catechism taught to children, which is practiced from childhood through adolescence to adulthood, leaves no alternative for a deliberate and conscientious effort at a religious ideological revolution. This article therefore suggests that deconstructing the structure of the polemic status of the prominent Christian and Islamic religious symbols through a planned inclusive pluralist religious curriculum will help refocus Nigeria's interreligious space. The effort is expected to yield the desired peace to facilitate the much-desired social, economic, and political development in Nigeria and beyond. Similarly, it is important to note that addressing intra-religious differences in the context of inclusive pluralism requires a long-term commitment and sustained efforts from individuals, communities, and institutions. By promoting dialogue, education, and mutual respect, societies can work towards fostering a climate of understanding and inclusivity, where religious violence based on sectarian differences can be mitigated.

**Funding:** This research received no external funding.

**Institutional Review Board Statement:** Not applicable.

**Informed Consent Statement:** Not applicable.

**Data Availability Statement:** All data from literary sources were duly acknowledged, no other researched data was sourced.

**Conflicts of Interest:** The author declares no conflict of interest.

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
