# Peer review of "A Deconstruction of the Cross and the Crescent for Inclusive Religious Pluralism between Muslims and Christians in Nigeria"

_religions, doi:10.3390/rel14060782_

Round 1
Reviewer 1 Report (Previous Reviewer 3)
I believe that the paper has improved following this round of revision.
Author Response
Thank you for the observations. A note is taken on required moderate English changes, and possible improvement on the paper.
Reviewer 2 Report (New Reviewer)
Review Report
General Comments
I. The submitted manuscript concerns a weighty empirical and theoretical issue. The language of the article is clear and elegant. Roughly put, the author’s main idea seems to be that we (or people in Nigeria) can overcome the destructive religious exclusivism and espouse a form of inclusive (or embracing) pluralism, chiefly by means of adopting a deconstructive way of seeing in which the meanings of religious symbols are considered to be constantly evolving and to have some opposing, contradictory senses and sentiments. This idea appears to be, to some extent, quite original and insightful.
II. Unfortunately, however, the article seems to have two fundamental difficulties. That is, from the theoretical point of view, the article does not seem to have a theoretical (or philosophical) rigor that can persuade professional scholars. On the other hand, from the practical point of view, it does not give us any concrete precepts or maxims that can impress some relevant people, such as “religious leaders”, “politicians”, “administrators”, “educators” and so on. The author of the paper says that he or she “demonstrated” his or her idea; but what the external observer sees is that, although the article deserves significant notice, the author is repeating his or her normative, stipulative claims without providing detailed and concrete expositions.
III. Thus seen, the article should be, theoretically, more rigorous and compelling; at the same time, it should be, practically, more serviceable and useful. Since the article has its own actual and potential merits, in order to develop those merits as fully as possible, it is suggested that the paper needs to be extensively revised even if it takes much time. And, when it is necessary, the length of the whole paper should be (or should be allowed to be) increased. Sections 2, 3, and 4 need to be condensed (some contents of those sections unnecessarily overlap). But sections 5 and 6 should be extended properly (even one or two new sections can be added). In that regard, the article’s main focus should be on the latter part of it (i.e., sections 5 and 6 etc.). The abstract of the paper should be re-written, placing some emphasis on its theoretical results and achievements.
Specific Comments
1. The author’s conceptions of inclusivism and pluralism should be re-examined more carefully. The author writes that “The third, religious inclusivism, recognizes some truth in every religion and that all religions lead to the same goal or ultimate reality” (lines 49-50). This is perplexing. Because it is the very idea of “religious pluralism” that the philosopher John Hick had repeatedly proposed in that area. (See Hick’s articles and books.) Paul Knitter (and many textbooks in the philosophy of religion) also adopted a similar view. (See Knitter’s book Introducing Theologies of Religions.) We need to note that according to what we call the mainstream view, inclusivism acknowledges the meaning or validity of other religions but attempts to include them in their own faith. Thus, the author’s theological topology of inclusivism and pluralism needs some textual evidence and/or theoretic justifications. Otherwise, the author’s view could be considered quite arbitrary or merely stipulative. (This is not to say that the author’s view is basically wrong.)
2. The article seems to need a more meticulous citing and referencing on the level of page numbers of cited texts. In addition to that, more detailed accounts will help the readers to understand the author’s position.
2.1 It seems that a main source of section 5 is Turner’ 2016 article. The author writes that “Christians and Muslims in Nigeria must move from mere inversion of the symbols’ outdated meanings to embrace the second stage of Derrida's deconstruction efforts, that is, searching for 'tensions, the contradictions, the heterogeneity within [the] corpus' (Caputo 1997)” (Lines 377-380). In the meantime, the Caputo quotation appears in Turner’s article in the same way. She writes: “Speaking at the Villanova Roundtable, Derrida described this as searching for the ‘tensions, the contradictions, the heterogeneity within [the] corpus” (Turner 2016; Also see her footnote 9). Those quotations made by the author and Turner are exactly the same. Of course, one can make the same quotation if it is needed. A worry is that the author might not have done a full research on the original texts of Derrida and other relevant philosophers.
Meanwhile, from the theoretical point of view, the author’s account of the distinction between the first and second stages of Derrida’s deconstruction concerning the meanings of religious symbols is somewhat unclear and insufficient. Probably this would be a more important point.
2.2 Let us also consider the author’s sentence “The projection of meaning depends on the inscribed differences in the structure of meaning (Line 361)”. This seems to be a result of paraphrasing Turner’s statement “The effect of the translation of thought into language is therefore to inscribe différance into the structure of meaning.” We know that there is a profound difference between the normal English term “difference” and Derrida’s theoretical term “différance”. (Note “é”.) Here it is not certain whether the author’s term “differences” is a typo or a theoretic paraphrase.
3. Overall, the reviewer has an impression that the author of the article deals with a very subtle, complicated problem in a naïve way. (The author’s language seems to be excellent, though.) Looked at this way, the reviewer raises the following foundational questions.
3.1 Unlike the author’s view, it may actually be doubtful whether the idea of deconstruction can be properly applicable to the issues of religion and religiosity. (If the research domain is politics, law, or ethics, this judgement could be different.) One feature of deconstructionism is that what we call the deconstructive process is a constantly ongoing process. This can help to dismantle some preconceived ideas and dogmatism. It can also assist us to see the new dimensions of religion or religiosity. On the other hand, however, it is not certain whether it can make us reach the final goal of religion or religiosity, i.e. realizing the ultimate truth or reality (whether it is salvation or enlightenment). If so, in our coping with religion, Derrida’s idea of deconstruction may turn out to be, after all, inadequate. Probably this has to do with the limits of deconstructionism in general, but the author seems to overlook that point.
3.2 The author, on the basis of Derrida’s ideas, seems to take it that the meanings of religious symbols in the established religions are evolving and changeable. But it is not the only way to solve the problem of religious exclusivism. Here, one should carefully note the term “meanings”. For the sake of argument, let us adopt the German logician Gottlob Frege’s view of meaning. (See Frege’s paper “The Thought”.) In his view, meaning has three aspects or types: meaning as referent (object), meaning as sense (concept), and meaning as subjective idea (psychological impression, feeling, or sentiment). In this regard, the meaning of a religious symbol taken as senses and subjective ideas can surely be evolving. But what about its ultimate referent or core senses? Should we say that the ultimate referent(s) of the symbol(s) is (are) also evolving and changeable? What could this mean in religious contexts? The point is that as far as the plausibility and validity of Derrida’s deconstructionism about the meanings of religious symbols are concerned, some more justifications are needed.
3.3 According to the author, for the purpose of avoiding religious conflicts and exclusivism, one can (or “should”), as an effective means, see and interpret the meanings of a religious symbol as contradictory. However, that sort of interpretation seems to be possible from the third-person, neutral point of view. More often than not, having a faith, individually or collectively, concerns first-person, subjective matters. If so, in what way and to what extent can a theorist persuade the sincere believers of a religion to adopt such a new, experimental way of thinking called “deconstruction”? Isn’t it, rather, too naïve a solution? Theoretically or practically, some more considerations are needed. Otherwise, it may turn out that the author is forcing us to take an extraordinary or, even, abnormal stance.
Exclusivism (in Nigeria) should be overcome. But it is not certain whether the task can be carried out via Derrida’s deconstructionism. What about, as an example, Hick’s pluralistic idea that all the different individual religions are distinct aspects of the one same ultimate reality? That seems to be a more accessible, intelligible, and serviceable solution to the problem, particularly in relation to Abrahamic religions. To repeat, some more detailed expositions and concrete elucidations of the author’s position are needed. (Cf. general comments II and III.)
Author Response
Thank you for the extensive attention to my article, Attention is being given to the observed details. Thanks
Below are the paragraph-by-paragraph response to the observations and corrections:
Review Report
General Comments
- The submitted manuscript concerns a weighty empirical and theoretical issue. The language of the article is clear and elegant. Roughly put, the author’s main idea seems to be that we (or people in Nigeria) can overcome the destructive religious exclusivism and espouse a form of inclusive (or embracing) pluralism, chiefly by means of adopting a deconstructive way of seeing in which the meanings of religious symbols are considered to be constantly evolving and to have some opposing, contradictory senses and sentiments. This idea appears to be, to some extent, quite original and insightful.
- The reviewer’s overview of the article is apt and in line with the author’s intention.
- Unfortunately, however, the article seems to have two fundamental difficulties. That is, from the theoretical point of view, the article does not seem to have a theoretical (or philosophical) rigor that can persuade professional scholars. On the other hand, from the practical point of view, it does not give us any concrete precepts or maxims that can impress some relevant people, such as “religious leaders”, “politicians”, “administrators”, “educators” and so on. The author of the paper says that he or she “demonstrated” his or her idea; but what the external observer sees is that, although the article deserves significant notice, the author is repeating his or her normative, stipulative claims without providing detailed and concrete expositions.
The three preliminary observations:
- The article does not seem to have a theoretical (or philosophical) rigor that can persuade professional scholars.
- It does not give us any concrete precepts or maxims that can impress some relevant people, such as “religious leaders”, “politicians”, “administrators”, “educators” and so on
- The author is repeating his or her normative, stipulative claims without providing detailed and concrete expositions.
Responses to the three preliminary observations were made:
- The first preliminary observation contradicts the reviewer’s later acknowledgement of an engagement of a philosophical theory. Nevertheless, the reviewer’s claim stated, it is difficult to measure the reviewer’s intended “theoretical (or philosophical)” assumptions since he/she did not identify what is in his/her mind. the assumed “rigor” that the article “does not seem to have” for the persuasion of “professional scholars” are unidentified.
- The article adopts a conceptual research approach which though deductive, is no less scientific than an inductive measure (empirical approach) that the reviewer would probably have loved the author to use to “impress” some people.
- There is no “normative, stipulative claim” in the article that is peculiar to the author as claimed by the reviewer. If the reference is to Inclusive Religious Pluralism as seemingly implied, the expression is not “stipulative” though it might not have been known to the reviewer. The expression is not part of the conventional straight jacket categorization of: inclusivism, exclusivism and pluralism. An objective insight into what is meant would have been most appreciated. But the reviewer neither explains nor identifies what he or she meant by the author’s “normative, stipulative claims” in respect of which the author did not provide detailed and concrete expositions. Nevertheless, the article is not about personal normative and stipulative claims, hence providing a detailed and concrete exposition on such a subjective stance would not be within the scope of the article.
III. Thus seen, the article should be, theoretically, more rigorous and compelling; at the same time, it should be, practically, more serviceable and useful. Since the article has its own actual and potential merits, in order to develop those merits as fully as possible, it is suggested that the paper needs to be extensively revised even if it takes much time. And, when it is necessary, the length of the whole paper should be (or should be allowed to be) increased. Sections 2, 3, and 4 need to be condensed (some contents of those sections unnecessarily overlap). But sections 5 and 6 should be extended properly (even one or two new sections can be added). In that regard, the article’s main focus should be on the latter part of it (i.e., sections 5 and 6 etc.). The abstract of the paper should be re-written, placing some emphasis on its theoretical results and achievements.
- The suggestion of condensing and expansion of isolated sections of the article, as well as the hint of rewriting the abstract does not agree with the other two reviewers and seems to be based on the current reviewer’s need for further understanding of the study point, scope, and methodology of the article.
Specific Comments
- The author’s conceptions of inclusivism and pluralism should be re-examined more carefully. The author writes that “The third, religious inclusivism, recognizes some truth in every religion and that all religions lead to the same goal or ultimate reality” (lines 49-50). This is perplexing. Because it is the very idea of “religious pluralism” that the philosopher John Hick had repeatedly proposed in that area. (See Hick’s articles and books.) Paul Knitter (and many textbooks in the philosophy of religion) also adopted a similar view. (See Knitter’s book Introducing Theologies of Religions.) We need to note that according to what we call the mainstream view, inclusivism acknowledges the meaning or validity of other religions but attempts to include them in their own faith. Thus, the author’s theological topology of inclusivism and pluralism needs some textual evidence and/or theoretic justifications. Otherwise, the author’s view could be considered quite arbitrary or merely stipulative. (This is not to say that the author’s view is basically wrong.)
- Thanks for the observation in line 50f. It has been amended. It was meant to read ‘religious inclusivism, recognizes some truth in every religion and that all religions lead to some form of goal or ultimate reality.’ However, it is important to note that the author neither generated an “arbitrary or merely stipulative” nor proposed a “perplexing” “topology” as the reviewer perceived. Inclusive religious pluralism may have not been known to the reviewer, but it is a category of discourse in interreligious relations which recognizes the gap between outright pluralism and inclusivism and provides a somewhat equilibrium approach to the beliefs of the religious other. For instance, while no Muslim would like to be baptized as an “anonymous Christian” (See Karl Rahner: Hearers of the Word, 1966; The Shape of the Church to Come, 1974; and Foundations of Christian Faith, 1978), Christians would not cede the belief in Christ as the way to salvation.
- Textual evidence abounds in contemporary discourses in interreligious/ interfaith dialogue. One such authority is the guest editor to the ongoing special edition of Religions, ("Religious Pluralism in the Contemporary Transformation Society"), Marinus Iwuchukwu. He is an authority on inclusive religious pluralism. (https://www.mdpi.com/journal/religions/special_issues/Pluralism_religions). He writes in the tradition of Jacques Dupuis, the father of inclusive religious pluralism in modern times.
- Some relevant “textual evidence” has been included in the revised manuscript (See ln 408).
- The article seems to need more meticulous citing and referencing on the level of page numbers of cited texts. In addition to that, more detailed accounts will help the readers to understand the author’s position.
- Noted with thanks.
- Attempt at “a more meticulous citing and referencing…” led to the use of Mendeley Reference Manager. More details have been included where deemed appropriate.
2.1 It seems that a main source of section 5 is Turner’ 2016 article. The author writes that “Christians and Muslims in Nigeria must move from mere inversion of the symbols’ outdated meanings to embrace the second stage of Derrida's deconstruction efforts, that is, searching for 'tensions, the contradictions, the heterogeneity within [the] corpus' (Caputo 1997)” (Lines 377-380). In the meantime, the Caputo quotation appears in Turner’s article in the same way. She writes: “Speaking at the Villanova Roundtable, Derrida described this as searching for the ‘tensions, the contradictions, the heterogeneity within [the] corpus” (Turner 2016; Also see her footnote 9). Those quotations made by the author and Turner are exactly the same. Of course, one can make the same quotation if it is needed. A worry is that the author might not have done a full research on the original texts of Derrida and other relevant philosophers.
Meanwhile, from the theoretical point of view, the author’s account of the distinction between the first and second stages of Derrida’s deconstruction concerning the meanings of religious symbols is somewhat unclear and insufficient. Probably this would be a more important point.
The suspicion of the author probably not doing full research belies an indictment from an unproven assumption. For the avoidance of doubt, both Turner, and Caputo papers were read and cited properly and accordingly. However, the latter has been dropped for a direct reference from Derrida’s work.
A more academic suggestion on “distinction between the first and second stages of Derrida’s deconstruction concerning the meanings of religious symbols” are taken and included in the revised work. Thanks.
2.2 Let us also consider the author’s sentence “The projection of meaning depends on the inscribed differences in the structure of meaning (Line 361)”. This seems to be a result of paraphrasing Turner’s statement “The effect of the translation of thought into language is therefore to inscribe différance into the structure of meaning.” We know that there is a profound difference between the normal English term “difference” and Derrida’s theoretical term “différance”. (Note “é”.) Here it is not certain whether the author’s term “differences” is a typo or a theoretic paraphrase.
The work of Turner was read by the author, as well as Derida’s Positions. Although the reviewer acknowledges a presumption of the author’s usage of Turner’s statement, it remains as such; the author is knowledgeable in the related field being discussed.
The use of the italicized differences in the article is deliberate and that is why it was italicized. The use was meant to retain Derrida’s intended usage of the English ‘difference’ and ‘deferral’. Note that Derrida coined the French "différance" (with an ‘a’) from "difference and deferral." His intention was also known to be an attempt to describe the way that meaning is always deferred or delayed. This intention and use of the word connote the English understanding. Note that words and concepts have varied meanings depending on the context in which they are used. Derrida’s use of "différance" is as a neologism from "difference" and "defer." He chose to spell the word with an "a" instead of an "e" to emphasize the difference between the two concepts. The author holds these contexts of usage in balance to present his position.
- Overall, the reviewer has an impression that the author of the article deals with a very subtle, complicated problem in a naïve way. (The author’s language seems to be excellent, though.)
The reviewer’s judgment of the author as naïve is personal and unsubstantiated. The reviewer apparently lacks competence in the area being discussed in the article, hence his/her perception of the author’s article as arising from “lack of experience, wisdom, or judgement” – a Dictionary meaning of naïve.
3.1 Unlike the author’s view, it may actually be doubtful whether the idea of deconstruction can be properly applicable to the issues of religion and religiosity. (If the research domain is politics, law, or ethics, this judgement could be different.) One feature of deconstructionism is that what we call the deconstructive process is a constantly ongoing process. This can help to dismantle some preconceived ideas and dogmatism. It can also assist us to see the new dimensions of religion or religiosity. On the other hand, however, it is not certain whether it can make us reach the final goal of religion or religiosity, i.e. realizing the ultimate truth or reality (whether it is salvation or enlightenment). If so, in our coping with religion, Derrida’s idea of deconstruction may turn out to be, after all, inadequate. Probably this has to do with the limits of deconstructionism in general, but the author seems to overlook that point.
Again, the competence of the reviewer in this area is questionable. Yes, deconstructionism is applicable to issues of religion and religiosity. Note that deconstructionism is a philosophical method that questions the assumptions underlying texts and discourses. The method is used to analyze religious texts, concepts, and traditions, to reveal their underlying contradictions and ambiguities. Derrida directly dealt with some levels of religious symbols in his Of Grammatology,
Of Spirit: Heidegger and the Question, Margins of Philosophy, and other writings. In the case of the article two religious symbols were analyzed to unveil a deeper understanding of the symbols. The idea is to understand that religious practices are not simply about following rules or repeating traditions. They are also about creating meaning and shaping identity, an idea that is in line with Derrida’s second stage.
For the avoidance of doubt, deconstructionism has been used to question the authority of religious texts, explore the relationship between religion and power, examine the role of gender and sexuality in religion, and analyze the relationship between religion and culture. The reviewer may find examples in the deconstructionist works of the following authors apart from Jacques Derrida; John Caputo, Mark C. Taylor, Catherine Keller, Harold Bloom, Paul de Man, Geoffrey Hartman, J. Hillis Miller, Barbara Johnson, Gayatri Spivak, and so on. The works of these authors and others have been influential in shaping the way that we think about religion today.
The author recognizes the fact that Deconstructionism is a controversial approach to issues of religion and religiosity, and he is sympathetic to the argument of some scholars that the approach is destructive and undermines the very foundations of religion. However, that position does not undermine the value of deconstructionism making the approach ready-to-hand in investigations beyond conventions.
3.2 The author, on the basis of Derrida’s ideas, seems to take it that the meanings of religious symbols in the established religions are evolving and changeable. But it is not the only way to solve the problem of religious exclusivism. Here, one should carefully note the term “meanings”. For the sake of argument, let us adopt the German logician Gottlob Frege’s view of meaning. (See Frege’s paper “The Thought”.) In his view, meaning has three aspects or types: meaning as referent (object), meaning as sense (concept), and meaning as subjective idea (psychological impression, feeling, or sentiment). In this regard, the meaning of a religious symbol taken as senses and subjective ideas can surely be evolving. But what about its ultimate referent or core senses? Should we say that the ultimate referent(s) of the symbol(s) is (are) also evolving and changeable? What could this mean in religious contexts? The point is that as far as the plausibility and validity of Derrida’s deconstructionism about the meanings of religious symbols are concerned, some more justifications are needed.
It is understood that the proposal of the author is not the only way to solve the problem of religious exclusivism. The author did not make that claim anywhere in the article but restricted himself to the defined scope. Nevertheless, the justification for the choice of Derrida’s deconstructionism has been addressed in the revised article, line 380f.
3.3 According to the author, for the purpose of avoiding religious conflicts and exclusivism, one can (or “should”), as an effective means, see and interpret the meanings of a religious symbol as contradictory. However, that sort of interpretation seems to be possible from the third-person, neutral point of view. More often than not, having a faith, individually or collectively, concerns first-person, subjective matters. If so, in what way and to what extent can a theorist persuade the sincere believers of a religion to adopt such a new, experimental way of thinking called “deconstruction”? Isn’t it, rather, too naïve a solution? Theoretically or practically, some more considerations are needed. Otherwise, it may turn out that the author is forcing us to take an extraordinary or, even, abnormal stance.
Academic works are rather suggestive (recommendations) rather than “forcing”. Again, the evidence of being personal rather than objective is evident in the allegation that “the author is forcing us to take an extraordinary or, even, abnormal stance.” There is nowhere in the article that the author attempts to “force” anyone including the reviewer and his/her assumed cohort (the us to whom he/she refers) to take what the reviewer unjustifiably referred to as “extraordinary or, even, abnormal stance.”
Research is conducted partly to shape opinions and positively impact society. A single individual may not assume to be everyone. Hence, the author fully agrees with the reviewer’s claim that “having a (sic) faith, individually or collectively, concerns first-person, subjective matters.” On that basis, conversions have been witnessed, as well as total rejection or modification of religious viewpoints based on a superior understanding of religious truth. On that basis, the author does not see how deconstructionism as an approach used by seasoned theologians and philosophers to unravel knotty religious issues is “too naïve a solution.”
Exclusivism (in Nigeria) should be overcome. But it is not certain whether the task can be carried out via Derrida’s deconstructionism. What about, as an example, Hick’s pluralistic idea that all the different individual religions are distinct aspects of the one same ultimate reality? That seems to be a more accessible, intelligible, and serviceable solution to the problem, particularly in relation to Abrahamic religions. To repeat, some more detailed expositions and concrete elucidations of the author’s position are needed. (Cf. general comments II and III.)
The personal position of the reviewer kept intruding on an objective assessment of the proposal in the article. This is betrayed by his/her concluding position which he/she claimed, “to be a more accessible, intelligible, and serviceable solution to the problem, particularly in relation to Abrahamic religions.” Note that Hick’s position is well known, but it lacks the balance of the differentiated implication of inclusivism and pluralism, hence the proposal of an inclusive religious pluralism which is apparently unknown to the reviewer.

Reviewer 3 Report (New Reviewer)

Author Response
Thanks for the observations, attention is being given to the review.
The observations and suggestions are well taken, thanks.
- Overall, I find the central argument persuasive, namely deconstructing the exclusive fixated understanding of the cross and the crescent. The author argues for a positive, non-violence meaning of the cross and the crescent. If the author has time, it might be worth reading Robert A. Dowd's Christianity, Islam, and Liberal Democracy: Lessons from Sub-Saharan Africa. (New York: Oxford University Press, 2015), 123-193. Dowd attributes the current peaceful coexistence between the different religious backgrounds in the south to the success of early inclusive interaction between the three religions- African Traditional, Christianity, and Islam.
Thanks for the suggestion of this book. I did have time ? A bit of it has been used to improve the inclusion of religious leaders in the work of incorporating inclusive pluralism in the affairs of the country: Lines 539 - 549
- Line 128-132: The author discusses the Muslim's need to liberate themselves from what they considered "the yoke of Euro-Christian values" Some Muslim radicals view Western education as evil. I was surprised that there was no mention of Boko Haram!
? the observation is taken. A mention of the group has been made in relation to the point of discussion. Lines 153 – 157.
- Line 87-88: The author discusses that some religious leaders are responsible for promoting religious exclusivism. It will be essential to address how inclusivism in religious education can be achieved when religious leaders claim so much power over religious education. For example, in 2016, the plan of inclusive religious education by the Ministry of Education in Ilorin state under the title Religion and National Values (R.N.V) was vehemently opposed by two religious’ leaders in Nigeria: Sa'ad Abubakar, the Sultan of Sokoto, and Archbishop Alfred Adewale Martins, the Catholic Archbishop of Lagos. These two religious leaders condemned the plan for an inclusive religious education and called on the Federal Government to maintain a separate Religious Education curriculum.
Revolutionary actions are not readily accepted, it might take some time to achieve inclusion of the proposed idea, however, dispositions of some religious leaders towards inclusive pluralism are now evident across the country. A few instances have been cited in the work: Lines 112 - 125
- Line 310: The author argued that the current curriculum is tailored for evangelization. What are some major themes in the curriculum, and what are some significant themes you want to see in an inclusive curriculum?
I have reflected the themes and my proposal in the revised work: Lines 338-347, and 353-357
- Islam, like Christianity, is not monolithic- different sects and denominations exist.
Often religious violence in Nigeria erupts due to intra-Muslim sectarian differences- inclusive religious education is unlikely to be achieved without an attempt to address intra-religious differences.
This observation is key to the topic of the article. I have taken the point to paper in pointing out that it is important to note that addressing intra-religious differences in the context of inclusive pluralism requires long-term commitment and sustained efforts from individuals, communities, and institutions. By promoting dialogue, education, and mutual respect, societies can work towards fostering a climate of understanding and inclusivity where religious violence based on sectarian differences can be mitigated. Line 558-563.
Round 2
Reviewer 2 Report (New Reviewer)
Comments
The submitted manuscript has been minimally revised. The author could have made a better revision. It seems that the author does not want to (or is not able to) do the task. In presenting and supporting the main idea of the paper, the author fails to see the difference between (a) theory-proposal, (b) theory-construction, and (c) theory-implementation. Recall that the article consists of two main parts: (1) unnecessarily long descriptions of a phenomenon in Nigeria (sections 1, 2, 3, and 4) and (2) short, insufficient theoretical treatments of the phenomenon (sections 5 and 6). The author seems to think that sections 5 and 6 pertain to (a), (b), and (c). That is the reason why the author’s result is naïve and incomplete. From the adequate philosophical point of view, it should be noted that what the author has actually done in the article is just “(a) theory-proposal”, but not “(b) theory-construction” or “(c) theory-implementation”. The paper becomes convincing and valuable when some more tasks associated with (b) and (c) are carried out.
* Some additional comments (blue-colored sentences) are given below.
Author’s Responses to Review Report
General Comments
I. The submitted manuscript concerns a weighty empirical and theoretical issue. The language of the article is clear and elegant. Roughly put, the author’s main idea seems to be that we (or people in Nigeria) can overcome the destructive religious exclusivism and espouse a form of inclusive (or embracing) pluralism, chiefly by means of adopting a deconstructive way of seeing in which the meanings of religious symbols are considered to be constantly evolving and to have some opposing, contradictory senses and sentiments. This idea appears to be, to some extent, quite original and insightful.
· The reviewer’s overview of the article is apt and in line with the author’s intention.
II. Unfortunately, however, the article seems to have two fundamental difficulties. That is, from the theoretical point of view, the article does not seem to have a theoretical (or philosophical) rigor that can persuade professional scholars. On the other hand, from the practical point of view, it does not give us any concrete precepts or maxims that can impress some relevant people, such as “religious leaders”, “politicians”, “administrators”, “educators” and so on. The author of the paper says that he or she “demonstrated” his or her idea; but what the external observer sees is that, although the article deserves significant notice, the author is repeating his or her normative, stipulative claims without providing detailed and concrete expositions.
The three preliminary observations:
1. The article does not seem to have a theoretical (or philosophical) rigor that can persuade professional scholars.
2. It does not give us any concrete precepts or maxims that can impress some relevant people, such as “religious leaders”, “politicians”, “administrators”, “educators” and so on
3. The author is repeating his or her normative, stipulative claims without providing detailed and concrete expositions.
Responses to the three preliminary observations were made:
1. The first preliminary observation contradicts the reviewer’s later acknowledgement of an engagement of a philosophical theory. Nevertheless, the reviewer’s claim stated, it is difficult to measure the reviewer’s intended “theoretical (or philosophical)” assumptions since he/she did not identify what is in his/her mind. the assumed “rigor” that the article “does not seem to have” for the persuasion of “professional scholars” are unidentified.
Exactly where does the reviewer make a contradiction? If one utilizes a philosophical idea to present and defend one’s thesis, he or she has to be equipped with theoretical rigor and details. That is, the burden of proof belongs to the author. The author’s paper is made up of colorful words and sentences, but it lacks convincing arguments.
2. The article adopts a conceptual research approach which though deductive, is no less scientific than an inductive measure (empirical approach) that the reviewer would probably have loved the author to use to “impress” some people.
3. There is no “normative, stipulative claim” in the article that is peculiar to the author as claimed by the reviewer. If the reference is to Inclusive Religious Pluralism as seemingly implied, the expression is not “stipulative” though it might not have been known to the reviewer. The expression is not part of the conventional straight jacket categorization of: inclusivism, exclusivism and pluralism. An objective insight into what is meant would have been most appreciated. But the reviewer neither explains nor identifies what he or she meant by the author’s “normative, stipulative claims” in respect of which the author did not provide detailed and concrete expositions. Nevertheless, the article is not about personal normative and stipulative claims, hence providing a detailed and concrete exposition on such a subjective stance would not be within the scope of the article.
The reviewer understands the author’s intention to use the term “inclusive.” The term may reflect the author’s theoretic concern. But the point we need to see is that the author’s way of using the term may rather bring about an unnecessary, negative connotation and misunderstanding. (Interestingly, the author does not note that the term “inclusive”, in this or that context, could have some opposing meanings delineated by the so-called Derrida’s deconstructionism.) And that is the reason why the author’s phraseology appears to be stipulative. Now, can we, then, also utilize the notion of “exclusive pluralism”? Yes, we can. But we should provide some conclusive provisos.
III. Thus seen, the article should be, theoretically, more rigorous and compelling; at the same time, it should be, practically, more serviceable and useful. Since the article has its own actual and potential merits, in order to develop those merits as fully as possible, it is suggested that the paper needs to be extensively revised even if it takes much time. And, when it is necessary, the length of the whole paper should be (or should be allowed to be) increased. Sections 2, 3, and 4 need to be condensed (some contents of those sections unnecessarily overlap). But sections 5 and 6 should be extended properly (even one or two new sections can be added). In that regard, the article’s main focus should be on the latter part of it (i.e., sections 5 and 6 etc.). The abstract of the paper should be re-written, placing some emphasis on its theoretical results and achievements.
· The suggestion of condensing and expansion of isolated sections of the article, as well as the hint of rewriting the abstract does not agree with the other two reviewers and seems to be based on the current reviewer’s need for further understanding of the study point, scope, and methodology of the article.
Probably those reviewers’ main areas may not be philosophy. Here, mentioning other reviewers’ views seems to be immature and inappropriate. That is to commit the logical fallacy of appealing to authority. Appeal to a good logic and theory.
Specific Comments
1. The author’s conceptions of inclusivism and pluralism should be re-examined more carefully. The author writes that “The third, religious inclusivism, recognizes some truth in every religion and that all religions lead to the same goal or ultimate reality” (lines 49-50). This is perplexing. Because it is the very idea of “religious pluralism” that the philosopher John Hick had repeatedly proposed in that area. (See Hick’s articles and books.) Paul Knitter (and many textbooks in the philosophy of religion) also adopted a similar view. (See Knitter’s book Introducing Theologies of Religions.) We need to note that according to what we call the mainstream view, inclusivism acknowledges the meaning or validity of other religions but attempts to include them in their own faith. Thus, the author’s theological topology of inclusivism and pluralism needs some textual evidence and/or theoretic justifications. Otherwise, the author’s view could be considered quite arbitrary or merely stipulative. (This is not to say that the author’s view is basically wrong.)
· Thanks for the observation in line 50f. It has been amended. It was meant to read ‘religious inclusivism, recognizes some truth in every religion and that all religions lead to some form of goal or ultimate reality.’ However, it is important to note that the author neither generated an “arbitrary or merely stipulative” nor proposed a “perplexing” “topology” as the reviewer perceived. Inclusive religious pluralism may have not been known to the reviewer, but it is a category of discourse in interreligious relations which recognizes the gap between outright pluralism and inclusivism and provides a somewhat equilibrium approach to the beliefs of the religious other. For instance, while no Muslim would like to be baptized as an “anonymous Christian” (See Karl Rahner: Hearers of the Word, 1966; The Shape of the Church to Come, 1974; and Foundations of Christian Faith, 1978), Christians would not cede the belief in Christ as the way to salvation.
· Textual evidence abounds in contemporary discourses in interreligious/ interfaith dialogue. One such authority is the guest editor to the ongoing special edition of Religions, ("Religious Pluralism in the Contemporary Transformation Society"), Marinus Iwuchukwu. He is an authority on inclusive religious pluralism. (https://www.mdpi.com/journal/religions/special_issues/Pluralism_religions). He writes in the tradition of Jacques Dupuis, the father of inclusive religious pluralism in modern times.
· Some relevant “textual evidence” has been included in the revised manuscript (See ln 408).
2. The article seems to need more meticulous citing and referencing on the level of page numbers of cited texts. In addition to that, more detailed accounts will help the readers to understand the author’s position.
· Noted with thanks.
· Attempt at “a more meticulous citing and referencing…” led to the use of Mendeley Reference Manager. More details have been included where deemed appropriate.
2.1 It seems that a main source of section 5 is Turner’ 2016 article. The author writes that “Christians and Muslims in Nigeria must move from mere inversion of the symbols’ outdated meanings to embrace the second stage of Derrida's deconstruction efforts, that is, searching for 'tensions, the contradictions, the heterogeneity within [the] corpus' (Caputo 1997)” (Lines 377-380). In the meantime, the Caputo quotation appears in Turner’s article in the same way. She writes: “Speaking at the Villanova Roundtable, Derrida described this as searching for the ‘tensions, the contradictions, the heterogeneity within [the] corpus” (Turner 2016; Also see her footnote 9). Those quotations made by the author and Turner are exactly the same. Of course, one can make the same quotation if it is needed. A worry is that the author might not have done a full research on the original texts of Derrida and other relevant philosophers.
Meanwhile, from the theoretical point of view, the author’s account of the distinction between the first and second stages of Derrida’s deconstruction concerning the meanings of religious symbols is somewhat unclear and insufficient. Probably this would be a more important point.
The suspicion of the author probably not doing full research belies an indictment from an unproven assumption. For the avoidance of doubt, both Turner, and Caputo papers were read and cited properly and accordingly. However, the latter has been dropped for a direct reference from Derrida’s work.
A more academic suggestion on “distinction between the first and second stages of Derrida’s deconstruction concerning the meanings of religious symbols” are taken and included in the revised work. Thanks.
2.2 Let us also consider the author’s sentence “The projection of meaning depends on the inscribed differences in the structure of meaning (Line 361)”. This seems to be a result of paraphrasing Turner’s statement “The effect of the translation of thought into language is therefore to inscribe différance into the structure of meaning.” We know that there is a profound difference between the normal English term “difference” and Derrida’s theoretical term “différance”. (Note “é”.) Here it is not certain whether the author’s term “differences” is a typo or a theoretic paraphrase.
The work of Turner was read by the author, as well as Derida’s Positions. Although the reviewer acknowledges a presumption of the author’s usage of Turner’s statement, it remains as such; the author is knowledgeable in the related field being discussed.
The use of the italicized differences in the article is deliberate and that is why it was italicized. The use was meant to retain Derrida’s intended usage of the English ‘difference’ and ‘deferral’. Note that Derrida coined the French "différance" (with an ‘a’) from "difference and deferral." His intention was also known to be an attempt to describe the way that meaning is always deferred or delayed. This intention and use of the word connote the English understanding. Note that words and concepts have varied meanings depending on the context in which they are used. Derrida’s use of "différance" is as a neologism from "difference" and "defer." He chose to spell the word with an "a" instead of an "e" to emphasize the difference between the two concepts. The author holds these contexts of usage in balance to present his position.
Note that in Turner’s article that the author paraphrased faithfully, Turner used the original French word “différance.” Here too, the point is that the author’s way of using the term “differences” may bring about not a conceptual assistance but a conceptual confusion or misunderstanding. It seems that the author never considers the readers.
3. Overall, the reviewer has an impression that the author of the article deals with a very subtle, complicated problem in a naïve way. (The author’s language seems to be excellent, though.)
The reviewer’s judgment of the author as naïve is personal and unsubstantiated. The reviewer apparently lacks competence in the area being discussed in the article, hence his/her perception of the author’s article as arising from “lack of experience, wisdom, or judgement” – a Dictionary meaning of naïve.
3.1 Unlike the author’s view, it may actually be doubtful whether the idea of deconstruction can be properly applicable to the issues of religion and religiosity. (If the research domain is politics, law, or ethics, this judgement could be different.) One feature of deconstructionism is that what we call the deconstructive process is a constantly ongoing process. This can help to dismantle some preconceived ideas and dogmatism. It can also assist us to see the new dimensions of religion or religiosity. On the other hand, however, it is not certain whether it can make us reach the final goal of religion or religiosity, i.e. realizing the ultimate truth or reality (whether it is salvation or enlightenment). If so, in our coping with religion, Derrida’s idea of deconstruction may turn out to be, after all, inadequate. Probably this has to do with the limits of deconstructionism in general, but the author seems to overlook that point.
Again, the competence of the reviewer in this area is questionable. Yes, deconstructionism is applicable to issues of religion and religiosity. Note that deconstructionism is a philosophical method that questions the assumptions underlying texts and discourses. The method is used to analyze religious texts, concepts, and traditions, to reveal their underlying contradictions and ambiguities. Derrida directly dealt with some levels of religious symbols in his Of Grammatology,
Of Spirit: Heidegger and the Question, Margins of Philosophy, and other writings. In the case of the article two religious symbols were analyzed to unveil a deeper understanding of the symbols. The idea is to understand that religious practices are not simply about following rules or repeating traditions. They are also about creating meaning and shaping identity, an idea that is in line with Derrida’s second stage.
For the avoidance of doubt, deconstructionism has been used to question the authority of religious texts, explore the relationship between religion and power, examine the role of gender and sexuality in religion, and analyze the relationship between religion and culture. The reviewer may find examples in the deconstructionist works of the following authors apart from Jacques Derrida; John Caputo, Mark C. Taylor, Catherine Keller, Harold Bloom, Paul de Man, Geoffrey Hartman, J. Hillis Miller, Barbara Johnson, Gayatri Spivak, and so on. The works of these authors and others have been influential in shaping the way that we think about religion today.
The author recognizes the fact that Deconstructionism is a controversial approach to issues of religion and religiosity, and he is sympathetic to the argument of some scholars that the approach is destructive and undermines the very foundations of religion. However, that position does not undermine the value of deconstructionism making the approach ready-to-hand in investigations beyond conventions.
Reread the original comments: “This can help to dismantle some preconceived ideas and dogmatism. It can also assist us to see the new dimensions of religion or religiosity. On the other hand, however, it is not certain whether it can make us reach the final goal of religion or religiosity, i.e. realizing the ultimate truth or reality (whether it is salvation or enlightenment).” They already contain what the author wants to say about deconstructionism.
3.2 The author, on the basis of Derrida’s ideas, seems to take it that the meanings of religious symbols in the established religions are evolving and changeable. But it is not the only way to solve the problem of religious exclusivism. Here, one should carefully note the term “meanings”. For the sake of argument, let us adopt the German logician Gottlob Frege’s view of meaning. (See Frege’s paper “The Thought”.) In his view, meaning has three aspects or types: meaning as referent (object), meaning as sense (concept), and meaning as subjective idea (psychological impression, feeling, or sentiment). In this regard, the meaning of a religious symbol taken as senses and subjective ideas can surely be evolving. But what about its ultimate referent or core senses? Should we say that the ultimate referent(s) of the symbol(s) is (are) also evolving and changeable? What could this mean in religious contexts? The point is that as far as the plausibility and validity of Derrida’s deconstructionism about the meanings of religious symbols are concerned, some more justifications are needed.
It is understood that the proposal of the author is not the only way to solve the problem of religious exclusivism. The author did not make that claim anywhere in the article but restricted himself to the defined scope. Nevertheless, the justification for the choice of Derrida’s deconstructionism has been addressed in the revised article, line 380f.
The author seems to avoid making a real answer to the problem above. It is the matter of deconstruction and ultimate religiosity.
3.3 According to the author, for the purpose of avoiding religious conflicts and exclusivism, one can (or “should”), as an effective means, see and interpret the meanings of a religious symbol as contradictory. However, that sort of interpretation seems to be possible from the third-person, neutral point of view. More often than not, having a faith, individually or collectively, concerns first-person, subjective matters. If so, in what way and to what extent can a theorist persuade the sincere believers of a religion to adopt such a new, experimental way of thinking called “deconstruction”? Isn’t it, rather, too naïve a solution? Theoretically or practically, some more considerations are needed. Otherwise, it may turn out that the author is forcing us to take an extraordinary or, even, abnormal stance.
Academic works are rather suggestive (recommendations) rather than “forcing”. Again, the evidence of being personal rather than objective is evident in the allegation that “the author is forcing us to take an extraordinary or, even, abnormal stance.” There is nowhere in the article that the author attempts to “force” anyone including the reviewer and his/her assumed cohort (the us to whom he/she refers) to take what the reviewer unjustifiably referred to as “extraordinary or, even, abnormal stance.”
Research is conducted partly to shape opinions and positively impact society. A single individual may not assume to be everyone. Hence, the author fully agrees with the reviewer’s claim that “having a (sic) faith, individually or collectively, concerns first-person, subjective matters.” On that basis, conversions have been witnessed, as well as total rejection or modification of religious viewpoints based on a superior understanding of religious truth. On that basis, the author does not see how deconstructionism as an approach used by seasoned theologians and philosophers to unravel knotty religious issues is “too naïve a solution.”
The reviewer is not saying that the author’s view is not totally wrong. It says that it requires more (philosophical) justifications.
Exclusivism (in Nigeria) should be overcome. But it is not certain whether the task can be carried out via Derrida’s deconstructionism. What about, as an example, Hick’s pluralistic idea that all the different individual religions are distinct aspects of the one same ultimate reality? That seems to be a more accessible, intelligible, and serviceable solution to the problem, particularly in relation to Abrahamic religions. To repeat, some more detailed expositions and concrete elucidations of the author’s position are needed. (Cf. general comments II and III.)
The personal position of the reviewer kept intruding on an objective assessment of the proposal in the article. This is betrayed by his/her concluding position which he/she claimed, “to be a more accessible, intelligible, and serviceable solution to the problem, particularly in relation to Abrahamic religions.” Note that Hick’s position is well known, but it lacks the balance of the differentiated implication of inclusivism and pluralism, hence the proposal of an inclusive religious pluralism which is apparently unknown to the reviewer.
The author does not recognize a possible danger associated with the phraseology “inclusive pluralism”. The author’s view seems to imply the following divisions: (i) inclusive pluralism based on Christianity (in Nigeria) and (ii) inclusive pluralism based on Muslim (in Nigeria). The point is that both (i) and (ii) could result in a form of exclusivism. Could there be a particular form of inclusive pluralism based on neutral deconstructionism? Does the Nigerian people follow it? Perhaps. However, more theoretical considerations (theory-construction and theory-implementation) are still needed.
Some typos appeared in the previous version of the paper are not still amended. For example, is the term “prima face” (line 429) correct? Where is the other side of the direct quotation marks (line 427, Turner quotation)? An evaluative claim about Encyclopedia Britannica (line 254) is somewhat strange and unnatural. It is the author himself/herself that has to make a good discussion in a scholarly paper.
Comments
The submitted manuscript has been minimally revised. The author could have made a better revision. It seems that the author does not want to (or is not able to) do the task. In presenting and supporting the main idea of the paper, the author fails to see the difference between (a) theory-proposal, (b) theory-construction, and (c) theory-implementation. Recall that the article consists of two main parts: (1) unnecessarily long descriptions of a phenomenon in Nigeria (sections 1, 2, 3, and 4) and (2) short, insufficient theoretical treatments of the phenomenon (sections 5 and 6). The author seems to think that sections 5 and 6 pertain to (a), (b), and (c). That is the reason why the author’s result is naïve and incomplete. From the adequate philosophical point of view, it should be noted that what the author has actually done in the article is just “(a) theory-proposal”, but not “(b) theory-construction” or “(c) theory-implementation”. The paper becomes convincing and valuable when some more tasks associated with (b) and (c) are carried out.
* Some additional comments (blue-colored sentences) are given below.
Author’s Responses to Review Report
General Comments
I. The submitted manuscript concerns a weighty empirical and theoretical issue. The language of the article is clear and elegant. Roughly put, the author’s main idea seems to be that we (or people in Nigeria) can overcome the destructive religious exclusivism and espouse a form of inclusive (or embracing) pluralism, chiefly by means of adopting a deconstructive way of seeing in which the meanings of religious symbols are considered to be constantly evolving and to have some opposing, contradictory senses and sentiments. This idea appears to be, to some extent, quite original and insightful.
· The reviewer’s overview of the article is apt and in line with the author’s intention.
II. Unfortunately, however, the article seems to have two fundamental difficulties. That is, from the theoretical point of view, the article does not seem to have a theoretical (or philosophical) rigor that can persuade professional scholars. On the other hand, from the practical point of view, it does not give us any concrete precepts or maxims that can impress some relevant people, such as “religious leaders”, “politicians”, “administrators”, “educators” and so on. The author of the paper says that he or she “demonstrated” his or her idea; but what the external observer sees is that, although the article deserves significant notice, the author is repeating his or her normative, stipulative claims without providing detailed and concrete expositions.
The three preliminary observations:
1. The article does not seem to have a theoretical (or philosophical) rigor that can persuade professional scholars.
2. It does not give us any concrete precepts or maxims that can impress some relevant people, such as “religious leaders”, “politicians”, “administrators”, “educators” and so on
3. The author is repeating his or her normative, stipulative claims without providing detailed and concrete expositions.
Responses to the three preliminary observations were made:
1. The first preliminary observation contradicts the reviewer’s later acknowledgement of an engagement of a philosophical theory. Nevertheless, the reviewer’s claim stated, it is difficult to measure the reviewer’s intended “theoretical (or philosophical)” assumptions since he/she did not identify what is in his/her mind. the assumed “rigor” that the article “does not seem to have” for the persuasion of “professional scholars” are unidentified.
Exactly where does the reviewer make a contradiction? If one utilizes a philosophical idea to present and defend one’s thesis, he or she has to be equipped with theoretical rigor and details. That is, the burden of proof belongs to the author. The author’s paper is made up of colorful words and sentences, but it lacks convincing arguments.
2. The article adopts a conceptual research approach which though deductive, is no less scientific than an inductive measure (empirical approach) that the reviewer would probably have loved the author to use to “impress” some people.
3. There is no “normative, stipulative claim” in the article that is peculiar to the author as claimed by the reviewer. If the reference is to Inclusive Religious Pluralism as seemingly implied, the expression is not “stipulative” though it might not have been known to the reviewer. The expression is not part of the conventional straight jacket categorization of: inclusivism, exclusivism and pluralism. An objective insight into what is meant would have been most appreciated. But the reviewer neither explains nor identifies what he or she meant by the author’s “normative, stipulative claims” in respect of which the author did not provide detailed and concrete expositions. Nevertheless, the article is not about personal normative and stipulative claims, hence providing a detailed and concrete exposition on such a subjective stance would not be within the scope of the article.
The reviewer understands the author’s intention to use the term “inclusive.” The term may reflect the author’s theoretic concern. But the point we need to see is that the author’s way of using the term may rather bring about an unnecessary, negative connotation and misunderstanding. (Interestingly, the author does not note that the term “inclusive”, in this or that context, could have some opposing meanings delineated by the so-called Derrida’s deconstructionism.) And that is the reason why the author’s phraseology appears to be stipulative. Now, can we, then, also utilize the notion of “exclusive pluralism”? Yes, we can. But we should provide some conclusive provisos.
III. Thus seen, the article should be, theoretically, more rigorous and compelling; at the same time, it should be, practically, more serviceable and useful. Since the article has its own actual and potential merits, in order to develop those merits as fully as possible, it is suggested that the paper needs to be extensively revised even if it takes much time. And, when it is necessary, the length of the whole paper should be (or should be allowed to be) increased. Sections 2, 3, and 4 need to be condensed (some contents of those sections unnecessarily overlap). But sections 5 and 6 should be extended properly (even one or two new sections can be added). In that regard, the article’s main focus should be on the latter part of it (i.e., sections 5 and 6 etc.). The abstract of the paper should be re-written, placing some emphasis on its theoretical results and achievements.
· The suggestion of condensing and expansion of isolated sections of the article, as well as the hint of rewriting the abstract does not agree with the other two reviewers and seems to be based on the current reviewer’s need for further understanding of the study point, scope, and methodology of the article.
Probably those reviewers’ main areas may not be philosophy. Here, mentioning other reviewers’ views seems to be immature and inappropriate. That is to commit the logical fallacy of appealing to authority. Appeal to a good logic and theory.
Specific Comments
1. The author’s conceptions of inclusivism and pluralism should be re-examined more carefully. The author writes that “The third, religious inclusivism, recognizes some truth in every religion and that all religions lead to the same goal or ultimate reality” (lines 49-50). This is perplexing. Because it is the very idea of “religious pluralism” that the philosopher John Hick had repeatedly proposed in that area. (See Hick’s articles and books.) Paul Knitter (and many textbooks in the philosophy of religion) also adopted a similar view. (See Knitter’s book Introducing Theologies of Religions.) We need to note that according to what we call the mainstream view, inclusivism acknowledges the meaning or validity of other religions but attempts to include them in their own faith. Thus, the author’s theological topology of inclusivism and pluralism needs some textual evidence and/or theoretic justifications. Otherwise, the author’s view could be considered quite arbitrary or merely stipulative. (This is not to say that the author’s view is basically wrong.)
· Thanks for the observation in line 50f. It has been amended. It was meant to read ‘religious inclusivism, recognizes some truth in every religion and that all religions lead to some form of goal or ultimate reality.’ However, it is important to note that the author neither generated an “arbitrary or merely stipulative” nor proposed a “perplexing” “topology” as the reviewer perceived. Inclusive religious pluralism may have not been known to the reviewer, but it is a category of discourse in interreligious relations which recognizes the gap between outright pluralism and inclusivism and provides a somewhat equilibrium approach to the beliefs of the religious other. For instance, while no Muslim would like to be baptized as an “anonymous Christian” (See Karl Rahner: Hearers of the Word, 1966; The Shape of the Church to Come, 1974; and Foundations of Christian Faith, 1978), Christians would not cede the belief in Christ as the way to salvation.
· Textual evidence abounds in contemporary discourses in interreligious/ interfaith dialogue. One such authority is the guest editor to the ongoing special edition of Religions, ("Religious Pluralism in the Contemporary Transformation Society"), Marinus Iwuchukwu. He is an authority on inclusive religious pluralism. (https://www.mdpi.com/journal/religions/special_issues/Pluralism_religions). He writes in the tradition of Jacques Dupuis, the father of inclusive religious pluralism in modern times.
· Some relevant “textual evidence” has been included in the revised manuscript (See ln 408).
2. The article seems to need more meticulous citing and referencing on the level of page numbers of cited texts. In addition to that, more detailed accounts will help the readers to understand the author’s position.
· Noted with thanks.
· Attempt at “a more meticulous citing and referencing…” led to the use of Mendeley Reference Manager. More details have been included where deemed appropriate.
2.1 It seems that a main source of section 5 is Turner’ 2016 article. The author writes that “Christians and Muslims in Nigeria must move from mere inversion of the symbols’ outdated meanings to embrace the second stage of Derrida's deconstruction efforts, that is, searching for 'tensions, the contradictions, the heterogeneity within [the] corpus' (Caputo 1997)” (Lines 377-380). In the meantime, the Caputo quotation appears in Turner’s article in the same way. She writes: “Speaking at the Villanova Roundtable, Derrida described this as searching for the ‘tensions, the contradictions, the heterogeneity within [the] corpus” (Turner 2016; Also see her footnote 9). Those quotations made by the author and Turner are exactly the same. Of course, one can make the same quotation if it is needed. A worry is that the author might not have done a full research on the original texts of Derrida and other relevant philosophers.
Meanwhile, from the theoretical point of view, the author’s account of the distinction between the first and second stages of Derrida’s deconstruction concerning the meanings of religious symbols is somewhat unclear and insufficient. Probably this would be a more important point.
The suspicion of the author probably not doing full research belies an indictment from an unproven assumption. For the avoidance of doubt, both Turner, and Caputo papers were read and cited properly and accordingly. However, the latter has been dropped for a direct reference from Derrida’s work.
A more academic suggestion on “distinction between the first and second stages of Derrida’s deconstruction concerning the meanings of religious symbols” are taken and included in the revised work. Thanks.
2.2 Let us also consider the author’s sentence “The projection of meaning depends on the inscribed differences in the structure of meaning (Line 361)”. This seems to be a result of paraphrasing Turner’s statement “The effect of the translation of thought into language is therefore to inscribe différance into the structure of meaning.” We know that there is a profound difference between the normal English term “difference” and Derrida’s theoretical term “différance”. (Note “é”.) Here it is not certain whether the author’s term “differences” is a typo or a theoretic paraphrase.
The work of Turner was read by the author, as well as Derida’s Positions. Although the reviewer acknowledges a presumption of the author’s usage of Turner’s statement, it remains as such; the author is knowledgeable in the related field being discussed.
The use of the italicized differences in the article is deliberate and that is why it was italicized. The use was meant to retain Derrida’s intended usage of the English ‘difference’ and ‘deferral’. Note that Derrida coined the French "différance" (with an ‘a’) from "difference and deferral." His intention was also known to be an attempt to describe the way that meaning is always deferred or delayed. This intention and use of the word connote the English understanding. Note that words and concepts have varied meanings depending on the context in which they are used. Derrida’s use of "différance" is as a neologism from "difference" and "defer." He chose to spell the word with an "a" instead of an "e" to emphasize the difference between the two concepts. The author holds these contexts of usage in balance to present his position.
Note that in Turner’s article that the author paraphrased faithfully, Turner used the original French word “différance.” Here too, the point is that the author’s way of using the term “differences” may bring about not a conceptual assistance but a conceptual confusion or misunderstanding. It seems that the author never considers the readers.
3. Overall, the reviewer has an impression that the author of the article deals with a very subtle, complicated problem in a naïve way. (The author’s language seems to be excellent, though.)
The reviewer’s judgment of the author as naïve is personal and unsubstantiated. The reviewer apparently lacks competence in the area being discussed in the article, hence his/her perception of the author’s article as arising from “lack of experience, wisdom, or judgement” – a Dictionary meaning of naïve.
3.1 Unlike the author’s view, it may actually be doubtful whether the idea of deconstruction can be properly applicable to the issues of religion and religiosity. (If the research domain is politics, law, or ethics, this judgement could be different.) One feature of deconstructionism is that what we call the deconstructive process is a constantly ongoing process. This can help to dismantle some preconceived ideas and dogmatism. It can also assist us to see the new dimensions of religion or religiosity. On the other hand, however, it is not certain whether it can make us reach the final goal of religion or religiosity, i.e. realizing the ultimate truth or reality (whether it is salvation or enlightenment). If so, in our coping with religion, Derrida’s idea of deconstruction may turn out to be, after all, inadequate. Probably this has to do with the limits of deconstructionism in general, but the author seems to overlook that point.
Again, the competence of the reviewer in this area is questionable. Yes, deconstructionism is applicable to issues of religion and religiosity. Note that deconstructionism is a philosophical method that questions the assumptions underlying texts and discourses. The method is used to analyze religious texts, concepts, and traditions, to reveal their underlying contradictions and ambiguities. Derrida directly dealt with some levels of religious symbols in his Of Grammatology,
Of Spirit: Heidegger and the Question, Margins of Philosophy, and other writings. In the case of the article two religious symbols were analyzed to unveil a deeper understanding of the symbols. The idea is to understand that religious practices are not simply about following rules or repeating traditions. They are also about creating meaning and shaping identity, an idea that is in line with Derrida’s second stage.
For the avoidance of doubt, deconstructionism has been used to question the authority of religious texts, explore the relationship between religion and power, examine the role of gender and sexuality in religion, and analyze the relationship between religion and culture. The reviewer may find examples in the deconstructionist works of the following authors apart from Jacques Derrida; John Caputo, Mark C. Taylor, Catherine Keller, Harold Bloom, Paul de Man, Geoffrey Hartman, J. Hillis Miller, Barbara Johnson, Gayatri Spivak, and so on. The works of these authors and others have been influential in shaping the way that we think about religion today.
The author recognizes the fact that Deconstructionism is a controversial approach to issues of religion and religiosity, and he is sympathetic to the argument of some scholars that the approach is destructive and undermines the very foundations of religion. However, that position does not undermine the value of deconstructionism making the approach ready-to-hand in investigations beyond conventions.
Reread the original comments: “This can help to dismantle some preconceived ideas and dogmatism. It can also assist us to see the new dimensions of religion or religiosity. On the other hand, however, it is not certain whether it can make us reach the final goal of religion or religiosity, i.e. realizing the ultimate truth or reality (whether it is salvation or enlightenment).” They already contain what the author wants to say about deconstructionism.
3.2 The author, on the basis of Derrida’s ideas, seems to take it that the meanings of religious symbols in the established religions are evolving and changeable. But it is not the only way to solve the problem of religious exclusivism. Here, one should carefully note the term “meanings”. For the sake of argument, let us adopt the German logician Gottlob Frege’s view of meaning. (See Frege’s paper “The Thought”.) In his view, meaning has three aspects or types: meaning as referent (object), meaning as sense (concept), and meaning as subjective idea (psychological impression, feeling, or sentiment). In this regard, the meaning of a religious symbol taken as senses and subjective ideas can surely be evolving. But what about its ultimate referent or core senses? Should we say that the ultimate referent(s) of the symbol(s) is (are) also evolving and changeable? What could this mean in religious contexts? The point is that as far as the plausibility and validity of Derrida’s deconstructionism about the meanings of religious symbols are concerned, some more justifications are needed.
It is understood that the proposal of the author is not the only way to solve the problem of religious exclusivism. The author did not make that claim anywhere in the article but restricted himself to the defined scope. Nevertheless, the justification for the choice of Derrida’s deconstructionism has been addressed in the revised article, line 380f.
The author seems to avoid making a real answer to the problem above. It is the matter of deconstruction and ultimate religiosity.
3.3 According to the author, for the purpose of avoiding religious conflicts and exclusivism, one can (or “should”), as an effective means, see and interpret the meanings of a religious symbol as contradictory. However, that sort of interpretation seems to be possible from the third-person, neutral point of view. More often than not, having a faith, individually or collectively, concerns first-person, subjective matters. If so, in what way and to what extent can a theorist persuade the sincere believers of a religion to adopt such a new, experimental way of thinking called “deconstruction”? Isn’t it, rather, too naïve a solution? Theoretically or practically, some more considerations are needed. Otherwise, it may turn out that the author is forcing us to take an extraordinary or, even, abnormal stance.
Academic works are rather suggestive (recommendations) rather than “forcing”. Again, the evidence of being personal rather than objective is evident in the allegation that “the author is forcing us to take an extraordinary or, even, abnormal stance.” There is nowhere in the article that the author attempts to “force” anyone including the reviewer and his/her assumed cohort (the us to whom he/she refers) to take what the reviewer unjustifiably referred to as “extraordinary or, even, abnormal stance.”
Research is conducted partly to shape opinions and positively impact society. A single individual may not assume to be everyone. Hence, the author fully agrees with the reviewer’s claim that “having a (sic) faith, individually or collectively, concerns first-person, subjective matters.” On that basis, conversions have been witnessed, as well as total rejection or modification of religious viewpoints based on a superior understanding of religious truth. On that basis, the author does not see how deconstructionism as an approach used by seasoned theologians and philosophers to unravel knotty religious issues is “too naïve a solution.”
The reviewer is not saying that the author’s view is not totally wrong. It says that it requires more (philosophical) justifications.
Exclusivism (in Nigeria) should be overcome. But it is not certain whether the task can be carried out via Derrida’s deconstructionism. What about, as an example, Hick’s pluralistic idea that all the different individual religions are distinct aspects of the one same ultimate reality? That seems to be a more accessible, intelligible, and serviceable solution to the problem, particularly in relation to Abrahamic religions. To repeat, some more detailed expositions and concrete elucidations of the author’s position are needed. (Cf. general comments II and III.)
The personal position of the reviewer kept intruding on an objective assessment of the proposal in the article. This is betrayed by his/her concluding position which he/she claimed, “to be a more accessible, intelligible, and serviceable solution to the problem, particularly in relation to Abrahamic religions.” Note that Hick’s position is well known, but it lacks the balance of the differentiated implication of inclusivism and pluralism, hence the proposal of an inclusive religious pluralism which is apparently unknown to the reviewer.
The author does not recognize a possible danger associated with the phraseology “inclusive pluralism”. The author’s view seems to imply the following divisions: (i) inclusive pluralism based on Christianity (in Nigeria) and (ii) inclusive pluralism based on Muslim (in Nigeria). The point is that both (i) and (ii) could result in a form of exclusivism. Could there be a particular form of inclusive pluralism based on neutral deconstructionism? Does the Nigerian people follow it? Perhaps. However, more theoretical considerations (theory-construction and theory-implementation) are still needed.
Some typos appeared in the previous version of the paper are not still amended. For example, is the term “prima face” (line 429) correct? Where is the other side of the direct quotation marks (line 427, Turner quotation)? An evaluative claim about Encyclopedia Britannica (line 254) is somewhat strange and unnatural. It is the author himself/herself that has to make a good discussion in a scholarly paper.
Comments
The submitted manuscript has been minimally revised. The author could have made a better revision. It seems that the author does not want to (or is not able to) do the task. In presenting and supporting the main idea of the paper, the author fails to see the difference between (a) theory-proposal, (b) theory-construction, and (c) theory-implementation. Recall that the article consists of two main parts: (1) unnecessarily long descriptions of a phenomenon in Nigeria (sections 1, 2, 3, and 4) and (2) short, insufficient theoretical treatments of the phenomenon (sections 5 and 6). The author seems to think that sections 5 and 6 pertain to (a), (b), and (c). That is the reason why the author’s result is naïve and incomplete. From the adequate philosophical point of view, it should be noted that what the author has actually done in the article is just “(a) theory-proposal”, but not “(b) theory-construction” or “(c) theory-implementation”. The paper becomes convincing and valuable when some more tasks associated with (b) and (c) are carried out.
* Some additional comments (blue-colored sentences) are given below.
Author’s Responses to Review Report
General Comments
I. The submitted manuscript concerns a weighty empirical and theoretical issue. The language of the article is clear and elegant. Roughly put, the author’s main idea seems to be that we (or people in Nigeria) can overcome the destructive religious exclusivism and espouse a form of inclusive (or embracing) pluralism, chiefly by means of adopting a deconstructive way of seeing in which the meanings of religious symbols are considered to be constantly evolving and to have some opposing, contradictory senses and sentiments. This idea appears to be, to some extent, quite original and insightful.
· The reviewer’s overview of the article is apt and in line with the author’s intention.
II. Unfortunately, however, the article seems to have two fundamental difficulties. That is, from the theoretical point of view, the article does not seem to have a theoretical (or philosophical) rigor that can persuade professional scholars. On the other hand, from the practical point of view, it does not give us any concrete precepts or maxims that can impress some relevant people, such as “religious leaders”, “politicians”, “administrators”, “educators” and so on. The author of the paper says that he or she “demonstrated” his or her idea; but what the external observer sees is that, although the article deserves significant notice, the author is repeating his or her normative, stipulative claims without providing detailed and concrete expositions.
The three preliminary observations:
1. The article does not seem to have a theoretical (or philosophical) rigor that can persuade professional scholars.
2. It does not give us any concrete precepts or maxims that can impress some relevant people, such as “religious leaders”, “politicians”, “administrators”, “educators” and so on
3. The author is repeating his or her normative, stipulative claims without providing detailed and concrete expositions.
Responses to the three preliminary observations were made:
1. The first preliminary observation contradicts the reviewer’s later acknowledgement of an engagement of a philosophical theory. Nevertheless, the reviewer’s claim stated, it is difficult to measure the reviewer’s intended “theoretical (or philosophical)” assumptions since he/she did not identify what is in his/her mind. the assumed “rigor” that the article “does not seem to have” for the persuasion of “professional scholars” are unidentified.
Exactly where does the reviewer make a contradiction? If one utilizes a philosophical idea to present and defend one’s thesis, he or she has to be equipped with theoretical rigor and details. That is, the burden of proof belongs to the author. The author’s paper is made up of colorful words and sentences, but it lacks convincing arguments.
2. The article adopts a conceptual research approach which though deductive, is no less scientific than an inductive measure (empirical approach) that the reviewer would probably have loved the author to use to “impress” some people.
3. There is no “normative, stipulative claim” in the article that is peculiar to the author as claimed by the reviewer. If the reference is to Inclusive Religious Pluralism as seemingly implied, the expression is not “stipulative” though it might not have been known to the reviewer. The expression is not part of the conventional straight jacket categorization of: inclusivism, exclusivism and pluralism. An objective insight into what is meant would have been most appreciated. But the reviewer neither explains nor identifies what he or she meant by the author’s “normative, stipulative claims” in respect of which the author did not provide detailed and concrete expositions. Nevertheless, the article is not about personal normative and stipulative claims, hence providing a detailed and concrete exposition on such a subjective stance would not be within the scope of the article.
The reviewer understands the author’s intention to use the term “inclusive.” The term may reflect the author’s theoretic concern. But the point we need to see is that the author’s way of using the term may rather bring about an unnecessary, negative connotation and misunderstanding. (Interestingly, the author does not note that the term “inclusive”, in this or that context, could have some opposing meanings delineated by the so-called Derrida’s deconstructionism.) And that is the reason why the author’s phraseology appears to be stipulative. Now, can we, then, also utilize the notion of “exclusive pluralism”? Yes, we can. But we should provide some conclusive provisos.
III. Thus seen, the article should be, theoretically, more rigorous and compelling; at the same time, it should be, practically, more serviceable and useful. Since the article has its own actual and potential merits, in order to develop those merits as fully as possible, it is suggested that the paper needs to be extensively revised even if it takes much time. And, when it is necessary, the length of the whole paper should be (or should be allowed to be) increased. Sections 2, 3, and 4 need to be condensed (some contents of those sections unnecessarily overlap). But sections 5 and 6 should be extended properly (even one or two new sections can be added). In that regard, the article’s main focus should be on the latter part of it (i.e., sections 5 and 6 etc.). The abstract of the paper should be re-written, placing some emphasis on its theoretical results and achievements.
· The suggestion of condensing and expansion of isolated sections of the article, as well as the hint of rewriting the abstract does not agree with the other two reviewers and seems to be based on the current reviewer’s need for further understanding of the study point, scope, and methodology of the article.
Probably those reviewers’ main areas may not be philosophy. Here, mentioning other reviewers’ views seems to be immature and inappropriate. That is to commit the logical fallacy of appealing to authority. Appeal to a good logic and theory.
Specific Comments
1. The author’s conceptions of inclusivism and pluralism should be re-examined more carefully. The author writes that “The third, religious inclusivism, recognizes some truth in every religion and that all religions lead to the same goal or ultimate reality” (lines 49-50). This is perplexing. Because it is the very idea of “religious pluralism” that the philosopher John Hick had repeatedly proposed in that area. (See Hick’s articles and books.) Paul Knitter (and many textbooks in the philosophy of religion) also adopted a similar view. (See Knitter’s book Introducing Theologies of Religions.) We need to note that according to what we call the mainstream view, inclusivism acknowledges the meaning or validity of other religions but attempts to include them in their own faith. Thus, the author’s theological topology of inclusivism and pluralism needs some textual evidence and/or theoretic justifications. Otherwise, the author’s view could be considered quite arbitrary or merely stipulative. (This is not to say that the author’s view is basically wrong.)
· Thanks for the observation in line 50f. It has been amended. It was meant to read ‘religious inclusivism, recognizes some truth in every religion and that all religions lead to some form of goal or ultimate reality.’ However, it is important to note that the author neither generated an “arbitrary or merely stipulative” nor proposed a “perplexing” “topology” as the reviewer perceived. Inclusive religious pluralism may have not been known to the reviewer, but it is a category of discourse in interreligious relations which recognizes the gap between outright pluralism and inclusivism and provides a somewhat equilibrium approach to the beliefs of the religious other. For instance, while no Muslim would like to be baptized as an “anonymous Christian” (See Karl Rahner: Hearers of the Word, 1966; The Shape of the Church to Come, 1974; and Foundations of Christian Faith, 1978), Christians would not cede the belief in Christ as the way to salvation.
· Textual evidence abounds in contemporary discourses in interreligious/ interfaith dialogue. One such authority is the guest editor to the ongoing special edition of Religions, ("Religious Pluralism in the Contemporary Transformation Society"), Marinus Iwuchukwu. He is an authority on inclusive religious pluralism. (https://www.mdpi.com/journal/religions/special_issues/Pluralism_religions). He writes in the tradition of Jacques Dupuis, the father of inclusive religious pluralism in modern times.
· Some relevant “textual evidence” has been included in the revised manuscript (See ln 408).
2. The article seems to need more meticulous citing and referencing on the level of page numbers of cited texts. In addition to that, more detailed accounts will help the readers to understand the author’s position.
· Noted with thanks.
· Attempt at “a more meticulous citing and referencing…” led to the use of Mendeley Reference Manager. More details have been included where deemed appropriate.
2.1 It seems that a main source of section 5 is Turner’ 2016 article. The author writes that “Christians and Muslims in Nigeria must move from mere inversion of the symbols’ outdated meanings to embrace the second stage of Derrida's deconstruction efforts, that is, searching for 'tensions, the contradictions, the heterogeneity within [the] corpus' (Caputo 1997)” (Lines 377-380). In the meantime, the Caputo quotation appears in Turner’s article in the same way. She writes: “Speaking at the Villanova Roundtable, Derrida described this as searching for the ‘tensions, the contradictions, the heterogeneity within [the] corpus” (Turner 2016; Also see her footnote 9). Those quotations made by the author and Turner are exactly the same. Of course, one can make the same quotation if it is needed. A worry is that the author might not have done a full research on the original texts of Derrida and other relevant philosophers.
Meanwhile, from the theoretical point of view, the author’s account of the distinction between the first and second stages of Derrida’s deconstruction concerning the meanings of religious symbols is somewhat unclear and insufficient. Probably this would be a more important point.
The suspicion of the author probably not doing full research belies an indictment from an unproven assumption. For the avoidance of doubt, both Turner, and Caputo papers were read and cited properly and accordingly. However, the latter has been dropped for a direct reference from Derrida’s work.
A more academic suggestion on “distinction between the first and second stages of Derrida’s deconstruction concerning the meanings of religious symbols” are taken and included in the revised work. Thanks.
2.2 Let us also consider the author’s sentence “The projection of meaning depends on the inscribed differences in the structure of meaning (Line 361)”. This seems to be a result of paraphrasing Turner’s statement “The effect of the translation of thought into language is therefore to inscribe différance into the structure of meaning.” We know that there is a profound difference between the normal English term “difference” and Derrida’s theoretical term “différance”. (Note “é”.) Here it is not certain whether the author’s term “differences” is a typo or a theoretic paraphrase.
The work of Turner was read by the author, as well as Derida’s Positions. Although the reviewer acknowledges a presumption of the author’s usage of Turner’s statement, it remains as such; the author is knowledgeable in the related field being discussed.
The use of the italicized differences in the article is deliberate and that is why it was italicized. The use was meant to retain Derrida’s intended usage of the English ‘difference’ and ‘deferral’. Note that Derrida coined the French "différance" (with an ‘a’) from "difference and deferral." His intention was also known to be an attempt to describe the way that meaning is always deferred or delayed. This intention and use of the word connote the English understanding. Note that words and concepts have varied meanings depending on the context in which they are used. Derrida’s use of "différance" is as a neologism from "difference" and "defer." He chose to spell the word with an "a" instead of an "e" to emphasize the difference between the two concepts. The author holds these contexts of usage in balance to present his position.
Note that in Turner’s article that the author paraphrased faithfully, Turner used the original French word “différance.” Here too, the point is that the author’s way of using the term “differences” may bring about not a conceptual assistance but a conceptual confusion or misunderstanding. It seems that the author never considers the readers.
3. Overall, the reviewer has an impression that the author of the article deals with a very subtle, complicated problem in a naïve way. (The author’s language seems to be excellent, though.)
The reviewer’s judgment of the author as naïve is personal and unsubstantiated. The reviewer apparently lacks competence in the area being discussed in the article, hence his/her perception of the author’s article as arising from “lack of experience, wisdom, or judgement” – a Dictionary meaning of naïve.
3.1 Unlike the author’s view, it may actually be doubtful whether the idea of deconstruction can be properly applicable to the issues of religion and religiosity. (If the research domain is politics, law, or ethics, this judgement could be different.) One feature of deconstructionism is that what we call the deconstructive process is a constantly ongoing process. This can help to dismantle some preconceived ideas and dogmatism. It can also assist us to see the new dimensions of religion or religiosity. On the other hand, however, it is not certain whether it can make us reach the final goal of religion or religiosity, i.e. realizing the ultimate truth or reality (whether it is salvation or enlightenment). If so, in our coping with religion, Derrida’s idea of deconstruction may turn out to be, after all, inadequate. Probably this has to do with the limits of deconstructionism in general, but the author seems to overlook that point.
Again, the competence of the reviewer in this area is questionable. Yes, deconstructionism is applicable to issues of religion and religiosity. Note that deconstructionism is a philosophical method that questions the assumptions underlying texts and discourses. The method is used to analyze religious texts, concepts, and traditions, to reveal their underlying contradictions and ambiguities. Derrida directly dealt with some levels of religious symbols in his Of Grammatology,
Of Spirit: Heidegger and the Question, Margins of Philosophy, and other writings. In the case of the article two religious symbols were analyzed to unveil a deeper understanding of the symbols. The idea is to understand that religious practices are not simply about following rules or repeating traditions. They are also about creating meaning and shaping identity, an idea that is in line with Derrida’s second stage.
For the avoidance of doubt, deconstructionism has been used to question the authority of religious texts, explore the relationship between religion and power, examine the role of gender and sexuality in religion, and analyze the relationship between religion and culture. The reviewer may find examples in the deconstructionist works of the following authors apart from Jacques Derrida; John Caputo, Mark C. Taylor, Catherine Keller, Harold Bloom, Paul de Man, Geoffrey Hartman, J. Hillis Miller, Barbara Johnson, Gayatri Spivak, and so on. The works of these authors and others have been influential in shaping the way that we think about religion today.
The author recognizes the fact that Deconstructionism is a controversial approach to issues of religion and religiosity, and he is sympathetic to the argument of some scholars that the approach is destructive and undermines the very foundations of religion. However, that position does not undermine the value of deconstructionism making the approach ready-to-hand in investigations beyond conventions.
Reread the original comments: “This can help to dismantle some preconceived ideas and dogmatism. It can also assist us to see the new dimensions of religion or religiosity. On the other hand, however, it is not certain whether it can make us reach the final goal of religion or religiosity, i.e. realizing the ultimate truth or reality (whether it is salvation or enlightenment).” They already contain what the author wants to say about deconstructionism.
3.2 The author, on the basis of Derrida’s ideas, seems to take it that the meanings of religious symbols in the established religions are evolving and changeable. But it is not the only way to solve the problem of religious exclusivism. Here, one should carefully note the term “meanings”. For the sake of argument, let us adopt the German logician Gottlob Frege’s view of meaning. (See Frege’s paper “The Thought”.) In his view, meaning has three aspects or types: meaning as referent (object), meaning as sense (concept), and meaning as subjective idea (psychological impression, feeling, or sentiment). In this regard, the meaning of a religious symbol taken as senses and subjective ideas can surely be evolving. But what about its ultimate referent or core senses? Should we say that the ultimate referent(s) of the symbol(s) is (are) also evolving and changeable? What could this mean in religious contexts? The point is that as far as the plausibility and validity of Derrida’s deconstructionism about the meanings of religious symbols are concerned, some more justifications are needed.
It is understood that the proposal of the author is not the only way to solve the problem of religious exclusivism. The author did not make that claim anywhere in the article but restricted himself to the defined scope. Nevertheless, the justification for the choice of Derrida’s deconstructionism has been addressed in the revised article, line 380f.
The author seems to avoid making a real answer to the problem above. It is the matter of deconstruction and ultimate religiosity.
3.3 According to the author, for the purpose of avoiding religious conflicts and exclusivism, one can (or “should”), as an effective means, see and interpret the meanings of a religious symbol as contradictory. However, that sort of interpretation seems to be possible from the third-person, neutral point of view. More often than not, having a faith, individually or collectively, concerns first-person, subjective matters. If so, in what way and to what extent can a theorist persuade the sincere believers of a religion to adopt such a new, experimental way of thinking called “deconstruction”? Isn’t it, rather, too naïve a solution? Theoretically or practically, some more considerations are needed. Otherwise, it may turn out that the author is forcing us to take an extraordinary or, even, abnormal stance.
Academic works are rather suggestive (recommendations) rather than “forcing”. Again, the evidence of being personal rather than objective is evident in the allegation that “the author is forcing us to take an extraordinary or, even, abnormal stance.” There is nowhere in the article that the author attempts to “force” anyone including the reviewer and his/her assumed cohort (the us to whom he/she refers) to take what the reviewer unjustifiably referred to as “extraordinary or, even, abnormal stance.”
Research is conducted partly to shape opinions and positively impact society. A single individual may not assume to be everyone. Hence, the author fully agrees with the reviewer’s claim that “having a (sic) faith, individually or collectively, concerns first-person, subjective matters.” On that basis, conversions have been witnessed, as well as total rejection or modification of religious viewpoints based on a superior understanding of religious truth. On that basis, the author does not see how deconstructionism as an approach used by seasoned theologians and philosophers to unravel knotty religious issues is “too naïve a solution.”
The reviewer is not saying that the author’s view is not totally wrong. It says that it requires more (philosophical) justifications.
Exclusivism (in Nigeria) should be overcome. But it is not certain whether the task can be carried out via Derrida’s deconstructionism. What about, as an example, Hick’s pluralistic idea that all the different individual religions are distinct aspects of the one same ultimate reality? That seems to be a more accessible, intelligible, and serviceable solution to the problem, particularly in relation to Abrahamic religions. To repeat, some more detailed expositions and concrete elucidations of the author’s position are needed. (Cf. general comments II and III.)
The personal position of the reviewer kept intruding on an objective assessment of the proposal in the article. This is betrayed by his/her concluding position which he/she claimed, “to be a more accessible, intelligible, and serviceable solution to the problem, particularly in relation to Abrahamic religions.” Note that Hick’s position is well known, but it lacks the balance of the differentiated implication of inclusivism and pluralism, hence the proposal of an inclusive religious pluralism which is apparently unknown to the reviewer.
The author does not recognize a possible danger associated with the phraseology “inclusive pluralism”. The author’s view seems to imply the following divisions: (i) inclusive pluralism based on Christianity (in Nigeria) and (ii) inclusive pluralism based on Muslim (in Nigeria). The point is that both (i) and (ii) could result in a form of exclusivism. Could there be a particular form of inclusive pluralism based on neutral deconstructionism? Does the Nigerian people follow it? Perhaps. However, more theoretical considerations (theory-construction and theory-implementation) are still needed.
Some typos appeared in the previous version of the paper are not still amended. For example, is the term “prima face” (line 429) correct? Where is the other side of the direct quotation marks (line 427, Turner quotation)? An evaluative claim about Encyclopedia Britannica (line 254) is somewhat strange and unnatural. It is the author himself/herself that has to make a good discussion in a scholarly paper.
Comments
The submitted manuscript has been minimally revised. The author could have made a better revision. It seems that the author does not want to (or is not able to) do the task. In presenting and supporting the main idea of the paper, the author fails to see the difference between (a) theory-proposal, (b) theory-construction, and (c) theory-implementation. Recall that the article consists of two main parts: (1) unnecessarily long descriptions of a phenomenon in Nigeria (sections 1, 2, 3, and 4) and (2) short, insufficient theoretical treatments of the phenomenon (sections 5 and 6). The author seems to think that sections 5 and 6 pertain to (a), (b), and (c). That is the reason why the author’s result is naïve and incomplete. From the adequate philosophical point of view, it should be noted that what the author has actually done in the article is just “(a) theory-proposal”, but not “(b) theory-construction” or “(c) theory-implementation”. The paper becomes convincing and valuable when some more tasks associated with (b) and (c) are carried out.
* Some additional comments (blue-colored sentences) are given below.
Author’s Responses to Review Report
General Comments
I. The submitted manuscript concerns a weighty empirical and theoretical issue. The language of the article is clear and elegant. Roughly put, the author’s main idea seems to be that we (or people in Nigeria) can overcome the destructive religious exclusivism and espouse a form of inclusive (or embracing) pluralism, chiefly by means of adopting a deconstructive way of seeing in which the meanings of religious symbols are considered to be constantly evolving and to have some opposing, contradictory senses and sentiments. This idea appears to be, to some extent, quite original and insightful.
· The reviewer’s overview of the article is apt and in line with the author’s intention.
II. Unfortunately, however, the article seems to have two fundamental difficulties. That is, from the theoretical point of view, the article does not seem to have a theoretical (or philosophical) rigor that can persuade professional scholars. On the other hand, from the practical point of view, it does not give us any concrete precepts or maxims that can impress some relevant people, such as “religious leaders”, “politicians”, “administrators”, “educators” and so on. The author of the paper says that he or she “demonstrated” his or her idea; but what the external observer sees is that, although the article deserves significant notice, the author is repeating his or her normative, stipulative claims without providing detailed and concrete expositions.
The three preliminary observations:
1. The article does not seem to have a theoretical (or philosophical) rigor that can persuade professional scholars.
2. It does not give us any concrete precepts or maxims that can impress some relevant people, such as “religious leaders”, “politicians”, “administrators”, “educators” and so on
3. The author is repeating his or her normative, stipulative claims without providing detailed and concrete expositions.
Responses to the three preliminary observations were made:
1. The first preliminary observation contradicts the reviewer’s later acknowledgement of an engagement of a philosophical theory. Nevertheless, the reviewer’s claim stated, it is difficult to measure the reviewer’s intended “theoretical (or philosophical)” assumptions since he/she did not identify what is in his/her mind. the assumed “rigor” that the article “does not seem to have” for the persuasion of “professional scholars” are unidentified.
Exactly where does the reviewer make a contradiction? If one utilizes a philosophical idea to present and defend one’s thesis, he or she has to be equipped with theoretical rigor and details. That is, the burden of proof belongs to the author. The author’s paper is made up of colorful words and sentences, but it lacks convincing arguments.
2. The article adopts a conceptual research approach which though deductive, is no less scientific than an inductive measure (empirical approach) that the reviewer would probably have loved the author to use to “impress” some people.
3. There is no “normative, stipulative claim” in the article that is peculiar to the author as claimed by the reviewer. If the reference is to Inclusive Religious Pluralism as seemingly implied, the expression is not “stipulative” though it might not have been known to the reviewer. The expression is not part of the conventional straight jacket categorization of: inclusivism, exclusivism and pluralism. An objective insight into what is meant would have been most appreciated. But the reviewer neither explains nor identifies what he or she meant by the author’s “normative, stipulative claims” in respect of which the author did not provide detailed and concrete expositions. Nevertheless, the article is not about personal normative and stipulative claims, hence providing a detailed and concrete exposition on such a subjective stance would not be within the scope of the article.
The reviewer understands the author’s intention to use the term “inclusive.” The term may reflect the author’s theoretic concern. But the point we need to see is that the author’s way of using the term may rather bring about an unnecessary, negative connotation and misunderstanding. (Interestingly, the author does not note that the term “inclusive”, in this or that context, could have some opposing meanings delineated by the so-called Derrida’s deconstructionism.) And that is the reason why the author’s phraseology appears to be stipulative. Now, can we, then, also utilize the notion of “exclusive pluralism”? Yes, we can. But we should provide some conclusive provisos.
III. Thus seen, the article should be, theoretically, more rigorous and compelling; at the same time, it should be, practically, more serviceable and useful. Since the article has its own actual and potential merits, in order to develop those merits as fully as possible, it is suggested that the paper needs to be extensively revised even if it takes much time. And, when it is necessary, the length of the whole paper should be (or should be allowed to be) increased. Sections 2, 3, and 4 need to be condensed (some contents of those sections unnecessarily overlap). But sections 5 and 6 should be extended properly (even one or two new sections can be added). In that regard, the article’s main focus should be on the latter part of it (i.e., sections 5 and 6 etc.). The abstract of the paper should be re-written, placing some emphasis on its theoretical results and achievements.
· The suggestion of condensing and expansion of isolated sections of the article, as well as the hint of rewriting the abstract does not agree with the other two reviewers and seems to be based on the current reviewer’s need for further understanding of the study point, scope, and methodology of the article.
Probably those reviewers’ main areas may not be philosophy. Here, mentioning other reviewers’ views seems to be immature and inappropriate. That is to commit the logical fallacy of appealing to authority. Appeal to a good logic and theory.
Specific Comments
1. The author’s conceptions of inclusivism and pluralism should be re-examined more carefully. The author writes that “The third, religious inclusivism, recognizes some truth in every religion and that all religions lead to the same goal or ultimate reality” (lines 49-50). This is perplexing. Because it is the very idea of “religious pluralism” that the philosopher John Hick had repeatedly proposed in that area. (See Hick’s articles and books.) Paul Knitter (and many textbooks in the philosophy of religion) also adopted a similar view. (See Knitter’s book Introducing Theologies of Religions.) We need to note that according to what we call the mainstream view, inclusivism acknowledges the meaning or validity of other religions but attempts to include them in their own faith. Thus, the author’s theological topology of inclusivism and pluralism needs some textual evidence and/or theoretic justifications. Otherwise, the author’s view could be considered quite arbitrary or merely stipulative. (This is not to say that the author’s view is basically wrong.)
· Thanks for the observation in line 50f. It has been amended. It was meant to read ‘religious inclusivism, recognizes some truth in every religion and that all religions lead to some form of goal or ultimate reality.’ However, it is important to note that the author neither generated an “arbitrary or merely stipulative” nor proposed a “perplexing” “topology” as the reviewer perceived. Inclusive religious pluralism may have not been known to the reviewer, but it is a category of discourse in interreligious relations which recognizes the gap between outright pluralism and inclusivism and provides a somewhat equilibrium approach to the beliefs of the religious other. For instance, while no Muslim would like to be baptized as an “anonymous Christian” (See Karl Rahner: Hearers of the Word, 1966; The Shape of the Church to Come, 1974; and Foundations of Christian Faith, 1978), Christians would not cede the belief in Christ as the way to salvation.
· Textual evidence abounds in contemporary discourses in interreligious/ interfaith dialogue. One such authority is the guest editor to the ongoing special edition of Religions, ("Religious Pluralism in the Contemporary Transformation Society"), Marinus Iwuchukwu. He is an authority on inclusive religious pluralism. (https://www.mdpi.com/journal/religions/special_issues/Pluralism_religions). He writes in the tradition of Jacques Dupuis, the father of inclusive religious pluralism in modern times.
· Some relevant “textual evidence” has been included in the revised manuscript (See ln 408).
2. The article seems to need more meticulous citing and referencing on the level of page numbers of cited texts. In addition to that, more detailed accounts will help the readers to understand the author’s position.
· Noted with thanks.
· Attempt at “a more meticulous citing and referencing…” led to the use of Mendeley Reference Manager. More details have been included where deemed appropriate.
2.1 It seems that a main source of section 5 is Turner’ 2016 article. The author writes that “Christians and Muslims in Nigeria must move from mere inversion of the symbols’ outdated meanings to embrace the second stage of Derrida's deconstruction efforts, that is, searching for 'tensions, the contradictions, the heterogeneity within [the] corpus' (Caputo 1997)” (Lines 377-380). In the meantime, the Caputo quotation appears in Turner’s article in the same way. She writes: “Speaking at the Villanova Roundtable, Derrida described this as searching for the ‘tensions, the contradictions, the heterogeneity within [the] corpus” (Turner 2016; Also see her footnote 9). Those quotations made by the author and Turner are exactly the same. Of course, one can make the same quotation if it is needed. A worry is that the author might not have done a full research on the original texts of Derrida and other relevant philosophers.
Meanwhile, from the theoretical point of view, the author’s account of the distinction between the first and second stages of Derrida’s deconstruction concerning the meanings of religious symbols is somewhat unclear and insufficient. Probably this would be a more important point.
The suspicion of the author probably not doing full research belies an indictment from an unproven assumption. For the avoidance of doubt, both Turner, and Caputo papers were read and cited properly and accordingly. However, the latter has been dropped for a direct reference from Derrida’s work.
A more academic suggestion on “distinction between the first and second stages of Derrida’s deconstruction concerning the meanings of religious symbols” are taken and included in the revised work. Thanks.
2.2 Let us also consider the author’s sentence “The projection of meaning depends on the inscribed differences in the structure of meaning (Line 361)”. This seems to be a result of paraphrasing Turner’s statement “The effect of the translation of thought into language is therefore to inscribe différance into the structure of meaning.” We know that there is a profound difference between the normal English term “difference” and Derrida’s theoretical term “différance”. (Note “é”.) Here it is not certain whether the author’s term “differences” is a typo or a theoretic paraphrase.
The work of Turner was read by the author, as well as Derida’s Positions. Although the reviewer acknowledges a presumption of the author’s usage of Turner’s statement, it remains as such; the author is knowledgeable in the related field being discussed.
The use of the italicized differences in the article is deliberate and that is why it was italicized. The use was meant to retain Derrida’s intended usage of the English ‘difference’ and ‘deferral’. Note that Derrida coined the French "différance" (with an ‘a’) from "difference and deferral." His intention was also known to be an attempt to describe the way that meaning is always deferred or delayed. This intention and use of the word connote the English understanding. Note that words and concepts have varied meanings depending on the context in which they are used. Derrida’s use of "différance" is as a neologism from "difference" and "defer." He chose to spell the word with an "a" instead of an "e" to emphasize the difference between the two concepts. The author holds these contexts of usage in balance to present his position.
Note that in Turner’s article that the author paraphrased faithfully, Turner used the original French word “différance.” Here too, the point is that the author’s way of using the term “differences” may bring about not a conceptual assistance but a conceptual confusion or misunderstanding. It seems that the author never considers the readers.
3. Overall, the reviewer has an impression that the author of the article deals with a very subtle, complicated problem in a naïve way. (The author’s language seems to be excellent, though.)
The reviewer’s judgment of the author as naïve is personal and unsubstantiated. The reviewer apparently lacks competence in the area being discussed in the article, hence his/her perception of the author’s article as arising from “lack of experience, wisdom, or judgement” – a Dictionary meaning of naïve.
3.1 Unlike the author’s view, it may actually be doubtful whether the idea of deconstruction can be properly applicable to the issues of religion and religiosity. (If the research domain is politics, law, or ethics, this judgement could be different.) One feature of deconstructionism is that what we call the deconstructive process is a constantly ongoing process. This can help to dismantle some preconceived ideas and dogmatism. It can also assist us to see the new dimensions of religion or religiosity. On the other hand, however, it is not certain whether it can make us reach the final goal of religion or religiosity, i.e. realizing the ultimate truth or reality (whether it is salvation or enlightenment). If so, in our coping with religion, Derrida’s idea of deconstruction may turn out to be, after all, inadequate. Probably this has to do with the limits of deconstructionism in general, but the author seems to overlook that point.
Again, the competence of the reviewer in this area is questionable. Yes, deconstructionism is applicable to issues of religion and religiosity. Note that deconstructionism is a philosophical method that questions the assumptions underlying texts and discourses. The method is used to analyze religious texts, concepts, and traditions, to reveal their underlying contradictions and ambiguities. Derrida directly dealt with some levels of religious symbols in his Of Grammatology,
Of Spirit: Heidegger and the Question, Margins of Philosophy, and other writings. In the case of the article two religious symbols were analyzed to unveil a deeper understanding of the symbols. The idea is to understand that religious practices are not simply about following rules or repeating traditions. They are also about creating meaning and shaping identity, an idea that is in line with Derrida’s second stage.
For the avoidance of doubt, deconstructionism has been used to question the authority of religious texts, explore the relationship between religion and power, examine the role of gender and sexuality in religion, and analyze the relationship between religion and culture. The reviewer may find examples in the deconstructionist works of the following authors apart from Jacques Derrida; John Caputo, Mark C. Taylor, Catherine Keller, Harold Bloom, Paul de Man, Geoffrey Hartman, J. Hillis Miller, Barbara Johnson, Gayatri Spivak, and so on. The works of these authors and others have been influential in shaping the way that we think about religion today.
The author recognizes the fact that Deconstructionism is a controversial approach to issues of religion and religiosity, and he is sympathetic to the argument of some scholars that the approach is destructive and undermines the very foundations of religion. However, that position does not undermine the value of deconstructionism making the approach ready-to-hand in investigations beyond conventions.
Reread the original comments: “This can help to dismantle some preconceived ideas and dogmatism. It can also assist us to see the new dimensions of religion or religiosity. On the other hand, however, it is not certain whether it can make us reach the final goal of religion or religiosity, i.e. realizing the ultimate truth or reality (whether it is salvation or enlightenment).” They already contain what the author wants to say about deconstructionism.
3.2 The author, on the basis of Derrida’s ideas, seems to take it that the meanings of religious symbols in the established religions are evolving and changeable. But it is not the only way to solve the problem of religious exclusivism. Here, one should carefully note the term “meanings”. For the sake of argument, let us adopt the German logician Gottlob Frege’s view of meaning. (See Frege’s paper “The Thought”.) In his view, meaning has three aspects or types: meaning as referent (object), meaning as sense (concept), and meaning as subjective idea (psychological impression, feeling, or sentiment). In this regard, the meaning of a religious symbol taken as senses and subjective ideas can surely be evolving. But what about its ultimate referent or core senses? Should we say that the ultimate referent(s) of the symbol(s) is (are) also evolving and changeable? What could this mean in religious contexts? The point is that as far as the plausibility and validity of Derrida’s deconstructionism about the meanings of religious symbols are concerned, some more justifications are needed.
It is understood that the proposal of the author is not the only way to solve the problem of religious exclusivism. The author did not make that claim anywhere in the article but restricted himself to the defined scope. Nevertheless, the justification for the choice of Derrida’s deconstructionism has been addressed in the revised article, line 380f.
The author seems to avoid making a real answer to the problem above. It is the matter of deconstruction and ultimate religiosity.
3.3 According to the author, for the purpose of avoiding religious conflicts and exclusivism, one can (or “should”), as an effective means, see and interpret the meanings of a religious symbol as contradictory. However, that sort of interpretation seems to be possible from the third-person, neutral point of view. More often than not, having a faith, individually or collectively, concerns first-person, subjective matters. If so, in what way and to what extent can a theorist persuade the sincere believers of a religion to adopt such a new, experimental way of thinking called “deconstruction”? Isn’t it, rather, too naïve a solution? Theoretically or practically, some more considerations are needed. Otherwise, it may turn out that the author is forcing us to take an extraordinary or, even, abnormal stance.
Academic works are rather suggestive (recommendations) rather than “forcing”. Again, the evidence of being personal rather than objective is evident in the allegation that “the author is forcing us to take an extraordinary or, even, abnormal stance.” There is nowhere in the article that the author attempts to “force” anyone including the reviewer and his/her assumed cohort (the us to whom he/she refers) to take what the reviewer unjustifiably referred to as “extraordinary or, even, abnormal stance.”
Research is conducted partly to shape opinions and positively impact society. A single individual may not assume to be everyone. Hence, the author fully agrees with the reviewer’s claim that “having a (sic) faith, individually or collectively, concerns first-person, subjective matters.” On that basis, conversions have been witnessed, as well as total rejection or modification of religious viewpoints based on a superior understanding of religious truth. On that basis, the author does not see how deconstructionism as an approach used by seasoned theologians and philosophers to unravel knotty religious issues is “too naïve a solution.”
The reviewer is not saying that the author’s view is not totally wrong. It says that it requires more (philosophical) justifications.
Exclusivism (in Nigeria) should be overcome. But it is not certain whether the task can be carried out via Derrida’s deconstructionism. What about, as an example, Hick’s pluralistic idea that all the different individual religions are distinct aspects of the one same ultimate reality? That seems to be a more accessible, intelligible, and serviceable solution to the problem, particularly in relation to Abrahamic religions. To repeat, some more detailed expositions and concrete elucidations of the author’s position are needed. (Cf. general comments II and III.)
The personal position of the reviewer kept intruding on an objective assessment of the proposal in the article. This is betrayed by his/her concluding position which he/she claimed, “to be a more accessible, intelligible, and serviceable solution to the problem, particularly in relation to Abrahamic religions.” Note that Hick’s position is well known, but it lacks the balance of the differentiated implication of inclusivism and pluralism, hence the proposal of an inclusive religious pluralism which is apparently unknown to the reviewer.
The author does not recognize a possible danger associated with the phraseology “inclusive pluralism”. The author’s view seems to imply the following divisions: (i) inclusive pluralism based on Christianity (in Nigeria) and (ii) inclusive pluralism based on Muslim (in Nigeria). The point is that both (i) and (ii) could result in a form of exclusivism. Could there be a particular form of inclusive pluralism based on neutral deconstructionism? Does the Nigerian people follow it? Perhaps. However, more theoretical considerations (theory-construction and theory-implementation) are still needed.
Some typos appeared in the previous version of the paper are not still amended. For example, is the term “prima face” (line 429) correct? Where is the other side of the direct quotation marks (line 427, Turner quotation)? An evaluative claim about Encyclopedia Britannica (line 254) is somewhat strange and unnatural. It is the author himself/herself that has to make a good discussion in a scholarly paper.

Author Response
Dear reviewer,
Thanks for helping to improve the quality of my paper. I have effected corrections in the suggested areas and offered explanations where requested (find attached my responses to your comments.
Thanks

This manuscript is a resubmission of an earlier submission. The following is a list of the peer review reports and author responses from that submission.
Round 1
Reviewer 1 Report
"A Deconstruction of the Cross and the Crescent for Inclusive Religious Pluralism between Muslims and Christians in Nigeria" attempts to described the role of the Cross and Crescent in religious crises within Nigeria. The article starts with a general introduction to the religious conflict between Christians and Muslims within Nigeria and the importance of this conflict with regards to political parties/power within the country.
While the historical description of the violence and conflict is present and adequate, but only addresses the goal of this article ("challenges and responds to the persistent antagonistic attitudes...") at the surface level.
There are a number of instances where assumptions are made without evidence. For instance, there is not evidence that people are 'comfortable with deteriorating values..." (Page 4, Line 172). Nor is there evidence regarding exclusive religious attitudes (Page 3, Line 174-177) or that the religions are 'systematically wired to make convert..." (Page 4, Line 181-184).
Rather, this article looks and acts more like a literature review or overview of the topic lacking the analyses/research to document the conclusions and arguments made throughout. I was hoping to see how the cross and crescent are resulting in violence today (e.g. surveys, interviews, etc. of citizens now). Or, and importantly, how the interactive religious model could work in practice.
Steps are documented, but again, these are not tested or addressed with a look at how it works within Nigeria beyond some general assumptions.
In addition to that, there are a number of areas that need to be revised or fixed. Here are some of the ones I noticed:
Page 2, Line 66: Why is Schools capitalized, but not high?
Page 3, Line 131: the author has 'is we discussed'.
Page 3, Line 141: the sentence "On the other hand," just ends.
Page 4, Line 161: the period after (2020) is not necessary.
Page 4, Line 167: The author uses Y-Studio and Y Studio, use consistent naming.
Page 4, Line 196: Forgot the s in Oppong's
Page 5, Line 224: no 's' needed in Muslims (Muslims monarchs)
Page 6, Line 302: oof needs to be of
Author Response
Thank you for the observations and suggestions.
The use of the "Cross" and the "Crescent" is meant to depict the long history of religious conflict, particularly between Christians and Muslims in the country, as well as to represent the tension and violence between the two groups. This intended meaning has been communicated in the revised version of the article. The symbols (as tangible Crosses and Flags) are not necessarily meant to be the cause of religious conflicts in the country. Although that did happen as narrated in the article, that is not always the case.
The article is not a product of empirical/quantitative research, hence the absence of your expected data, and quantitative approach.
Identified 'assumptions made without evidence' have been substantiated.
Observations on typos and spellings have been corrected
Reviewer 2 Report
The objective of the study („This article attempts to answer the question of how the exclusive religious disposition underlining most instances of religious crises in the country may be addressed”) is not achieved.
The study does not reflect a concrete scientific research. Rather it is a descriptive approach without substance.
The conceptual constructs that the author mentions: "inclusive religious pluralism", "exclusive religious disposition" are not explained.
Remarks such as "Jihad (as war against Christians) or crusade (as war against Muslims)" are unfounded.
Author Response
Observations made from the contents and construct of the articles have been addressed. Part of the objectives of the study is to answer the question of how the exclusive religious disposition underlines most instances of religious crises in the country, that has been revisited.
The study is qualitative in approach and deployed a historical highlight as background to what the paper intends to achieve.
The conceptual constructs mentioned: "inclusive religious pluralism", "exclusive religious disposition" have been explained.
Although remarks such as "Jihad (as war against Christians) or crusade (as war against Muslims)" which were adjudged unfounded, find expression in the work of Moosa, M. (2008). 2. Islam and Christianity: Jihad and Holy War, the expressions have been deleted.
Further attention was paid to other details in the work.
Thanks for the observation, and suggestions, and for improving the work.
Reviewer 3 Report
The article is well-written and engaging to read, and I commend the authors' efforts, recommend a revision, and offer my suggestions.
The central argument needs to be strengthened, deepened, and streamlined by using the two primary recommendations below: (1) colonial legacies as a critical lens; and (2) multi-dimensional subscriptions of three religions. Apart from the above two core issues, the conclusion is also too short.
Please take a look at the attachment for details.

Author Response
Thank you for the suggestions on specific areas to improve the article. While a substantial part of the suggestions has been incorporated, I felt an introduction of aspects of the Traditional African Spiritual System (TASS) would distract the focus of the article from the Christian and Islamic symbols on which the scope rests.
Other suggestions have been noted and taken care of

Round 2
Reviewer 2 Report
To this revised form of the article, I dont have any comments.
Reviewer 3 Report
The point-to-point response to my previous comment needed to be attached to the submission. It is a vital aspect of the review process.
· Outstanding issues
You have not provided a point-to-point response to my previous comments.
· Three ways to revise a paper
Ideally, after reviewers' reports and recommendations, there are three broad ways authors can revise their manuscript for the second or third round of review to create more transparency.
[1] One way of doing it is to revise the paper following the list of reviewers' comments to improve it.
[2] Another way of doing it is to decline to revise the paper in line with all reviewers' comments by giving reasons to support that decision concerning each component of the reviewers’ comments.
[3] Furthermore, another way can combine routes (1) and (2) listed above in revising a paper as per reviewers' comments to create more transparency.
· Either way, a further significant revision is warranted.
The critical point is that the peer-review process gives authors a unique opportunity to have a three-way dialogue with external examiners or referees (involving journal editors).
Acceptance or further revision of papers following a peer-review process should be contingent upon the logical point-to-point responses to previous comments from external referees.
Otherwise, the purpose of the whole review process is defeated and meaningless.
Author Response
A point-by-point response to the to 3rd Reviewer’s suggestion has been done as follows:
I acknowledge the commendation and recommendation of the reviewer on the being “well-written and engaging to read”, and “efforts.” I have worked on suggestions to enhance the focus of the paper, which is on the interaction of two specific religions in Nigeria (Islam and Christianity) and specific attitudes of their adherents to certain symbols and beliefs.
A step-by-step attempt at recommendations and suggestions has been integrated where possible. They are as follows:
On strengthening, deepening, and streamlining the paper’s argument using two primary recommendations of (1) colonial legacies as a critical lens, and (2) multi-dimensional subscriptions of three religions: The section on Nigeria's ever-present presence of strained interreligious uneasiness, and thoughts on religious dichotomy and conflicts between modern Christianity and Islam in Nigeria have been collapsed into an extensive discussion in An inherited imbalance of relio-political dichotomy in Nigeria.
On integrating suggested sources: “Tweets and Reactions: Revealing the Geographies of Cybercrime Perpetrators and the North-South Divide. Cyberpsychology, Behavior, and Social Networking”; "Where Is the Money? The Intersectionality of the Spirit World and the Acquisition of Wealth”; and “How God became a Nigerian: Religious impulse and the unfolding of a nation.” The sources have been well interrogated for use in the article. The sources are found to be well written in their context but not all have much to do with the theme of the article.
Tweets and Reactions: Revealing the Geographies of Cybercrime Perpetrators and the North-South Divide focused on the production and prosecution of cybercrime by means of a postcolonial perspective, to shed light on the legacies of British colonial efforts in Nigeria. The article has some bearing on the religious divide in the country and has been appropriated in the article.
Where Is the Money? The Intersectionality of the Spirit World and the Acquisition of Wealth addressed “the ways in which local worldviews on wealth acquisition give rise to contemporary manifestations of spirituality in cyberspace.” The contents of the article do not directly or significantly interrogate the article, which specifically centers on inclusive religious pluralism between Islam and Christianity.
African Traditions in the Study of Religion in Africa: Emerging Trends, Indigenous Spirituality and the Interface with Other World Religions resonates with Where Is the Money? The Intersectionality of the Spirit World and the Acquisition of Wealth. Although the book addresses religion headlong, it addresses new perspectives and trends in the emergence of African Traditions in the study of religions in Africa. Its focus in that direction is on African traditions in the study of religion in Africa and the new African diaspora. It does not have much to do with the issues engaged in the paper.
Than you for the input at making the paper better
